EMBO
Molecular Medicine

# Menin-regulated Pbk controls high fat diet-induced compensatory beta cell proliferation

Jian Ma[1,2] , Bowen Xing[1], Yan Cao[1,2], Xin He[1] , Kate E Bennett[3], Chao Tong[4], Chiying An[1,2], Taylor Hojnacki[1], Zijie Feng[1], Sunbin Deng[5], Sunbin Ling[1], Gengchen Xie[1], Yuan Wu[1], Yue Ren[6], Ming Yu[2], Bryson W Katona[1,3], Hongzhe Li[6], Ali Naji[2] & Xianxin Hua[1,2,*]

## Abstract

Pancreatic beta cells undergo compensatory proliferation in the early phase of type 2 diabetes. While pathways such as FoxM1 are involved in regulating compensatory beta cell proliferation, given the lack of therapeutics effectively targeting beta cell proliferation, other targetable pathways need to be identified. Herein, we show that Pbk, a serine/threonine protein kinase, is essential for high fat diet (HFD)-induced beta cell proliferation *in vivo* using a Pbk kinase deficiency knock-in mouse model. Mechanistically, JunD recruits menin and HDAC3 complex to the *Pbk* promoter to reduce histone H3 acetylation, leading to epigenetic repression of Pbk expression. Moreover, menin inhibitor (MI) disrupts the menin–JunD interaction and augments *Pbk* transcription. Importantly, MI administration increases beta cell proliferation, ameliorating hyperglycemia, and impaired glucose tolerance (IGT) in HFD-induced diabetic mice. Notably, Pbk is required for the MI-induced beta cell proliferation and improvement of IGT. Together, these results demonstrate the repressive role of the menin/JunD/Pbk axis in regulating HFD-induced compensatory beta cell proliferation and pharmacologically regulating this axis may serve as a novel strategy for type 2 diabetes therapy.

**Keywords** beta cell; compensatory proliferation; diabetes; menin; Pbk
**Subject Categories** Digestive System; Metabolism

## Introduction

Beta cells in pancreatic islets undergo compensatory proliferation in the early phase of obesity and high fat diet (HFD)-induced type 2 diabetes (T2D); however, long-term stress in beta cells induced by these conditions eventually results in beta cell death and a reduced number of functional beta cells (Finegood *et al,* 2001; Hanley *et al,* 2010; Saisho *et al,* 2010; Linnemann *et al,* 2014; Wang *et al,* 2015). As such, there has been considerable interest in understanding how compensatory beta cell proliferation is regulated, and how the underlying mechanism can be further explored for improving diabetes treatment (Kulkarni *et al,* 2004; Linnemann *et al,* 2014; El Ouaamari *et al,* 2016). A number of factors such as transcription factor FoxM1, incretin hormone GLP-1, liver-derived protease inhibitor serpinB1, RNA splicing processing protein Argonaute2, and others are involved in the compensatory beta cell proliferation (Baggio & Drucker, 2007; Dai *et al,* 2017; El Ouaamari *et al,* 2016; Kawamori *et al,* 2017; Ning *et al,* 2006; Tattikota *et al,* 2014; Yamamoto *et al,* 2017), but other unknown pathways that regulate this process remain to be identified.

Pbk, a serine/threonine protein kinase, facilitates cell cycle control and mitotic progression (Abe *et al,* 2000; Gaudet *et al,* 2000; Fujibuchi *et al,* 2005). Pbk mRNA is detected in limited human tissue types and is most abundant in testis, placenta, thymus, and activated lymphoid cells (Abe *et al,* 2000; Zhu *et al,* 2007). Normally, Pbk expression is suppressed in non-transformed cells from differentiated tissues including pancreatic beta cells (Ayllon & O'Connor, 2007; Joel *et al,* 2015; Herbert *et al,* 2018). Furthermore, it remains unclear whether Pbk plays any role in beta cell proliferation.

Menin, which is encoded by the *MEN1* gene, is an established key regulator of beta cell mass, as ablation of the *Men1* gene leads to increased beta cell proliferation in mice (Crabtree *et al,* 2001; Bertolino *et al,* 2003; Schnepp *et al,* 2006; Karnik *et al,* 2007). *Men1* gene deletion also reverses pre-existing hyperglycemia in high fat diet (HFD)-induced diabetic mice as well as in obese (db/db) mice (Schnepp *et al,* 2006; Yang *et al,* 2010a; Yang *et al,* 2010b) and ameliorates gestational diabetes in pregnant mice (Karnik *et al,* 2007). Menin interacts with various protein partners and regulates beta cell homeostasis via multiple pathways including regulation of gene transcription and cell proliferation (Agarwal *et al,* 1999; Jin *et al,* 2003; Hughes *et al,* 2004; Milne *et al,* 2005; Grembecka *et al,*

1 Department of Cancer Biology, Abramson Family Cancer Research Institute, University of Pennsylvania Perelman School of Medicine, Philadelphia, PA, USA
2 Institute for Diabetes, Obesity, and Metabolism, University of Pennsylvania Perelman School of Medicine, Philadelphia, PA, USA
3 Division of Gastroenterology, University of Pennsylvania Perelman School of Medicine, Philadelphia, PA, USA
4 Department of Biology, University of Pennsylvania, Philadelphia, PA, USA
5 Department of Chemistry, University of Pennsylvania, Philadelphia, PA, USA
6 Department of Biochemistry and Biophysics, University of Pennsylvania Perelman School of Medicine, Philadelphia, PA, USA
*Corresponding author. Tel: +1 215 746 5565; E-mail: huax@pennmedicine.upenn.edu

2012; Gurung *et al*, 2013; Matkar *et al*, 2013). Among the different interacting partners, the menin–MLL interaction plays a suppressing role in islet cells by driving expression of cell cycle inhibitors p27 and p18 (Karnik *et al*, 2005; Milne *et al*, 2005). In addition, menin also binds JunD, an AP-1 family transcription factor (Agarwal *et al*, 1999; Kim *et al*, 2003). X-ray crystallographic studies showed that menin's structure harbors a deep central pocket that binds MLL or JunD (Huang *et al*, 2012; Shi *et al*, 2012). Small molecule menin inhibitors (MIs), including MI-463 and MI-503, were developed to target this pocket and block the menin–MLL interaction, and MIs are effective at suppressing MLL-fusion protein-induced leukemia *in vivo* (Borkin *et al*, 2015).

Pbk is upregulated in *Men1*-excised islets in mouse models (Yang *et al*, 2010a) and in human pancreatic neuroendocrine tumor (PNET) tissues with *MEN1* loss-of-function mutations (Jiang *et al*, 2014). These findings prompted us to investigate whether menin regulates Pbk expression as well as the mechanism of this regulation and also whether Pbk is crucial for the beta cell compensatory proliferation induced by stressors such as HFD.

Herein, for the first time we demonstrated the crucial role of Pbk in regulating HFD-induced beta cell compensatory proliferation and uncovered the repressive function of the menin/JunD/Pbk axis in regulating compensatory beta cell proliferation. We showed that administration of MI, which can interrupt the menin/JunD/Pbk axis, upregulates Pbk expression, improves compensatory beta cell proliferation, and potently improves hyperglycemia in HFD-induced diabetic mice. These studies may accelerate the development of new types of diabetes therapies directed at promoting Pbk-dependent beta cell proliferation.

# Results

## Pbk is upregulated in islets of mice with high fat diet (HFD)-induced obesity and crucial for beta cell proliferation

Recent gene expression profile assays in pancreatic beta cells or islets from HFD rodent models show increased Pbk expression, opening a question about the possible role of enhanced Pbk expression in regulating HFD-induced pancreatic beta cell compensatory proliferation. Analysis of Pbk expression profile in normal human and mouse tissues shows that Pbk protein/mRNA expression is mainly restricted to bone marrow, testis, and the gastrointestinal tract (Appendix Fig S1A and B). Moreover, further analysis of the mRNA expression profile of mice on HFD vs control mice (Malpique *et al*, 2014; Dusaulcy *et al*, 2019) indicates that Pbk is upregulated in the islets of mice on HFD (Fig 1A and Appendix Fig S1C). To confirm these findings, we performed qRT–PCR with the isolated islets from the C57BL/6 mice on HFD or chow diet, showing that Pbk mRNA level was higher in islets from HFD-induced obese mice than

**Figure 1. Pbk kinase expression is upregulated in pancreatic beta cells of HFD-induced diabetic mice with beta cell compensatory proliferation.**

A Volcano plot showing the fold change (*y*-axis) versus adjusted (adj.) *P* value (*x*-axis) of the beta cell transcriptomes between chow and HFD-fed mice (16 weeks). Genes highlighted in red or green are based on the thresholds of $Log_2$ fold change > 1 and adj. *P* value < 0.01 (two-tailed paired Student's *t*-test). Genes with > 10-fold upregulated expression are labeled with their name. Pbk gene expression is indicated by the arrow. RNA-seq data were available from Dusaulcy *et al* (2019) (https://doi.org/10.1371/journal.pone.0213299.t004).

B qPCR for detecting Pbk mRNA level in different metabolically active organs from DIO and control mice (*n* = 3 per group). \*\**P* = 0.0029 (islets), \*\*\**P* = 0.0007 (muscle) (two-tailed unpaired Student's *t*-test). ns, not statistically significant difference.

C PBK mRNA level comparison in human islets from obese (BMI > 30) or lean donors (BMI < 25) (*n* = 3 per group). \*\*P = 0.0261 (two-tailed unpaired Student's *t*-test).

D Representative images of insulin (green) and Pbk (red, indicated by white arrow) double immuno-staining in pancreas of DIO mice with 12 weeks HFD feeding (60% fat diet, *n* = 4) or age-matched control mice (10% fat diet, *n* = 4). Nuclei were labeled by DAPI (blue). Islet area was circled with white dashed line. Scale bar: 50 μm.

E Quantification of the percentage of Pbk-positive beta cells. Four mice for each group, and 10 islet images per mouse were analyzed. \*P = 0.0150 (two-tailed unpaired Student's *t*-test).

F WB data showed an upregulation of Pbk expression level in islets of mice on HFD compared with that of control lean mice.

G Beta cell mass comparison between DIO and control mice (*n* = 4 per group). \*P = 0.0419 (two-tailed unpaired Student's *t*-test).

H Representative images of insulin (green) and BrdU (red, indicated by white arrow) double immuno-staining in pancreas of DIO mice with 12 weeks HFD feeding (60% fat diet, *n* = 4) or age-matched control mice (10% fat diet, *n* = 4). Nuclei were labeled by DAPI (blue). Islet area was circled with white dashed line. Scale bar: 50 μm.

I Quantification of the percentage of BrdU positive-β-cells. Four mice for each group, and 5–10 islet images per mouse were analyzed. \*P = 0.0230 (two-tailed unpaired Student's *t*-test).

J The representative image of co-staining of Pbk and BrdU in pancreatic sections. Islet area was circled with white dashed line. Pbk and BrdU co-staining are denoted by white arrows. Scale bar: 50 μm.

K Quantification of the percentage with pbk and BrdU co-staining cells among total Pbk positive cells in HFD-fed mouse islets. The mean value of four mice from 5 sections for each mouse were presented.

L Pbk KD in INS-1 cells suppresses cell growth. Western blot data showed the decreased Pbk expression level in INS-1 cells with Pbk-targeted shRNA transduction. Cell growth curve was from three independent experiments (*n* = 3). \*\*\**P* < 0.0001 (Vector:shRNA-#1), \*P = 0.0181 (Vector:shRNA-#5), (Two-way ANOVA).

M The role of ectopic expression of Pbk-WT and Pbk-Mut on INS-1 cell growth. Western blot data showed the overexpression of V5-flaged wild-type and mutant Pbk in INS-1 cells. Cell growth curve was from three independent experiments (*n* = 3). \*\*P = 0.0069 (Two-way ANOVA). ns, not statistically significant difference.

N Cell cycle analysis via PI staining followed by flow cytometry for the cells with Pbk overexpression. PIME cells steadily transduced with vector, Pbk-WT, and Pbk-Mut expression plasmids.

O Purified recombinant His-Erk2 protein incubated with the purified PBK in the presence or absence of ATP (100 μM) or menin inhibitor (1 μM or 0.1 μM), as indicated. After SDS–PAGE, the phosphorylation of Erk2 was detected with the pERK1/2 antibody. Anti-ERK1/2 antibody showed that equal amount of Erk2 was used for the kinase assay.

P Purified recombinant His-Erk2 proteins were incubated with IP-ed Pbk from PIME cells (WT or menin-/-) for the kinase activity assay. The phosphorylation of Erk2 was detected with the anti-pERK1/2 antibody. The Western blot incubated with the anti-ERK1/2 antibody showed that an equal amount of Erk2 was loaded for the kinase assay.

Data information: Data are represented as mean ± SEM.

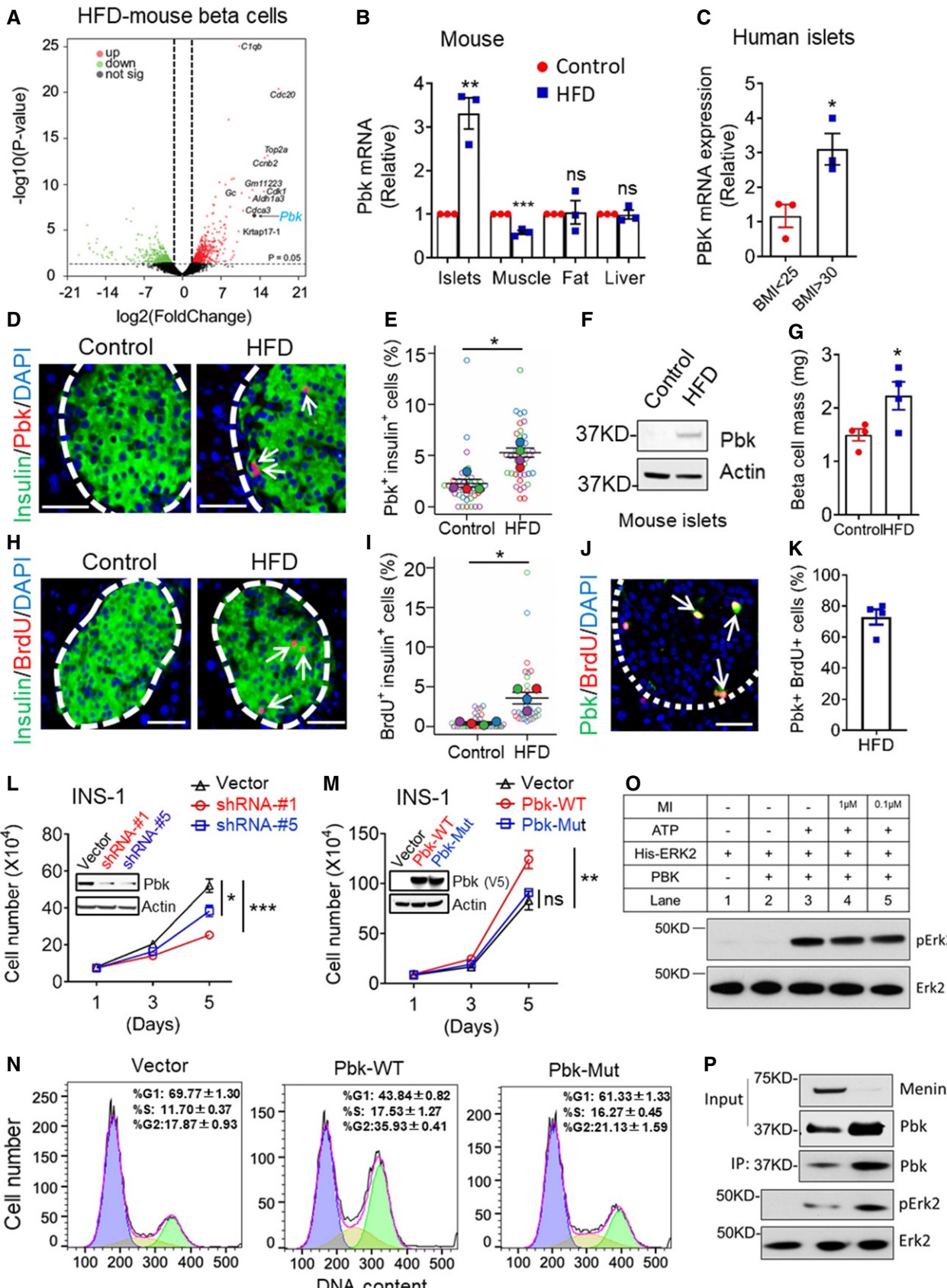

**Figure 1.**

that from the control lean mice. In contrast, Pbk mRNA was not significantly increased in other metabolically active tissues such as fat and liver, and was even downregulated in muscle (Fig 1B).

Moreover, increased PBK expression was also identified in human islets from obese donors (BMI > 30) as compared to the lean donors (BMI < 25) (Fig 1C, Appendix Table S1). Upregulated Pbk protein was confirmed via detecting an increased number of Pbk/insulin co-stained beta cells from the HFD-induced obese mice (Fig 1D and E) and an increased Pbk protein expression in isolated islet samples of HFD-induced obese mice relative to the control lean mice (Fig 1F). Consistent with increased Pbk expression, beta cell mass and BrdU uptake in beta cells of HFD-induced obese mice were also increased (Fig 1G–I). Further, Pbk and BrdU co-localized, with approximately 70% of Pbk-positive cells also showing BrdU positivity, indicating Pbk-mediated cell proliferation is cell-autonomous (Fig 1J and K). Together, these results support that HFD upregulates Pbk expression selectively in islets of mice, correlating with enhanced beta cell proliferation.

Next, to elucidate the relationship between upregulated Pbk expression and beta cell proliferation, we showed that shRNA-mediated Pbk knockdown (KD) suppressed growth of INS-1 cells, a rat insulinoma-derived cell line (Fig 1L), whereas ectopic expression of Pbk promoted INS-1 cell growth (Fig 1M). Notably, a $K^{64}K^{65}$ to AA mutation in Pbk, which deactivates catalytic activity of Pbk (Gaudet et al, 2000), does not promote INS-1 cell growth as seen with wild-type Pbk (Fig 1M). To further determine whether Pbk expression is crucial for cell proliferation, we treated Pbk overexpressing and control PIME cells, a mouse islet-derived cell line with an inducible floxed *Men1* gene (Appendix Fig S2), with Pbk-specific inhibitor OTS514 (Ikeda et al, 2016). The results showed that ectopic expression of Pbk increased the sensitivity of PIME cells to the Pbk inhibitor (Fig EV1A). Furthermore, upregulated Pbk expression drove more cells into G2/M phase (Fig 1N), increased phosphorylation of Erk1/2 and JunD, and upregulated CcnB1 expression (Fig EV1B and C). Consistently, the Pbk-mutant cells had fewer cells in G2/M phase (Fig 1N). Taken together, these results indicate

that Pbk kinase activity plays a crucial role in promoting beta cell proliferation.

To explore the substrate of Pbk kinase, we performed the *in vitro* kinases assay with purified Pbk and Erk2, both with or without ATP. The results show that incubation of Pbk with Erk2 in the presence of ATP indeed induced Erk2 phosphorylation (Fig 1O, lane 3). In contrast and as expected, removal of either Pbk or ATP from the assay abolished Erk2 phosphorylation (Fig 1O, lane 1 and 2, respectively). On the other hand, addition of various concentrations of menin inhibitor (MI) (Borkin et al, 2015), which was used to induce menin-dependent Pbk upregulation in the subsequent *in vitro* or *in vivo* experiments, did not affect the Pbk-mediated Erk2 phosphorylation (Fig 1O, lanes 4–5), indicating MI does not physically impact Pbk kinase activity. As a control, the amount of purified Erk2 was comparable among each of the reactions (Fig 1O, lanes 1–5, lower panel). In addition, we further examined in cells whether upregulated Pbk could also lead to improved kinase activity. To this end, we immunoprecipitated (IP-ed) Pbk from wild-type PIME cells and menin KO PIME cells. First, we confirmed the upregulation of Pbk expression in menin KO PIME cells compared with the control PIME cells (Fig 1P, Input Pbk). Then, we IP-ed Pbk from the same amount of PIME cell lysates (with or without menin) and performed *in vitro* kinase assays showing that larger amounts of Pbk induced higher level of Erk2 phosphorylation (Fig 1P). Together, these results indicate that Pbk can phosphorylate Erk2.

## Generation of Pbk kinase-inactivated mice and the essential role of Pbk in maintaining normal glucose tolerance

To investigate the *in vivo* role of Pbk in regulating HFD-induced beta cell proliferation, we generated Pbk kinase inactivation mutant Knockin (KI) mice by introducing KI of Pbk with a $K^{64}K^{65} \rightarrow AA$ mutation (referred to as Pbk$^{KI/KI}$ hereafter) (Fig 2A and B). Homozygous Pbk$^{KI/KI}$ mice were identified using restriction fragment length polymorphism (RFLP) analysis as well as DNA sequencing (Fig 2B–D).

---

**Figure 2.   Pbk kinase activity is crucial for HFD-induced beta cell compensatory proliferation.**

A    A diagram for generation of Pbk $K^{64}K^{65} \rightarrow$ AA KI mice using CRISPR/Cas9 system.

B    Target region in exon5 of murine *Pbk* locus. Mutant nucleotides for inducing KK → AA mutation were highlighted with blue. Two mismatched bases with silence mutations that generate a novel Mwol site are labeled in yellow. The amplicon and enzyme cutting size are indicated.

C    PCR genotyping of tail genomic DNA for Pbk$^{KI/KI}$ mice using RFLP. Genomic fragments from exon 5 were amplified by PCR and digested with Mwol.

D    DNA sequence confirming the mutation from K64K65 to AA in Pbk$^{KI/KI}$ mice from the tail genomic DNA compared with Pbk$^{WT/WT}$ mice.

E    IPGTT (glucose at 2 g/kg of body weight, i.p.) on 11-week-old male Pbk$^{KI/KI}$ mice and Pbk$^{WT/WT}$ mice ($n = 4$ for each group). All mice were starved overnight (16 h) before testing.

F    Islet number in each pancreas section on 11-week-old male Pbk$^{KI/KI}$ and Pbk$^{WT/WT}$ mice ($n = 4$ for each group). *$P = 0.0443$ (two-tailed unpaired Student's $t$-test).

G    Beta cell mass comparison between 11-week-old male Pbk$^{KI/KI}$ and Pbk$^{WT/WT}$ mice ($n = 4$ for each group). **$P = 0.0015$ (two-tailed unpaired Student's $t$-test).

H    *Ex vitro* perfusion studies on the islets isolated from 11-week-old male Pbk$^{KI/KI}$ and Pbk$^{WT/WT}$ mice ($n = 3$ for each group).

I    HFD feeding schedule for Pbk$^{KI/KI}$ and Pbk$^{WT/WT}$ mice. 7-week-old male Pbk$^{KI/KI}$ and Pbk$^{WT/WT}$ mice were fed with HFD ($n = 8$ for each group) for 12 weeks.

J    Persistent measurement of GTT for Pbk$^{KI/KI}$ or Pbk$^{WT/WT}$ mice on HFD ($n = 4$ for each group) or chow diet ($n = 2$ for each group) at various times, i.e., 0, 3$^{rd}$, 5$^{th}$, 8$^{th}$, and 12$^{th}$ week during HFD feeding. The AUC data of GTT at various time points were shown. *$P = 0.0294$, (Two-way ANOVA). ns, not statistically significant difference.

K, L    Representative images of double immuno-staining for insulin (green) and BrdU (red) of pancreas from Pbk$^{WT/WT}$ (K) or Pbk$^{KI/KI}$ (L) mice with 5 weeks HFD feeding. Islet area was circled with white dashed line. Nuclei were labeled by DAPI (blue). Scale bar: 50 μm.

M    Quantification of BrdU and insulin dual positive cells ($n = 4$ per group). Four mice for each group, and 5–10 islet images per mouse were analyzed. *$P = 0.0220$ (two-tailed unpaired Student's $t$-test).

N    Beta cell mass comparison between Pbk$^{KI/KI}$ and Pbk$^{WT/WT}$ mice ($n = 4$ per group). **$P = 0.0010$ (two-tailed unpaired Student's $t$-test).

O    Quantification of Pbk expression positive beta cells. Four mice for each group, 10 islet images per mouse were analyzed. ns, not statistically significant difference (unpaired two-tailed Student's $t$-test).

Data information: Data are represented as mean ± SEM.

---

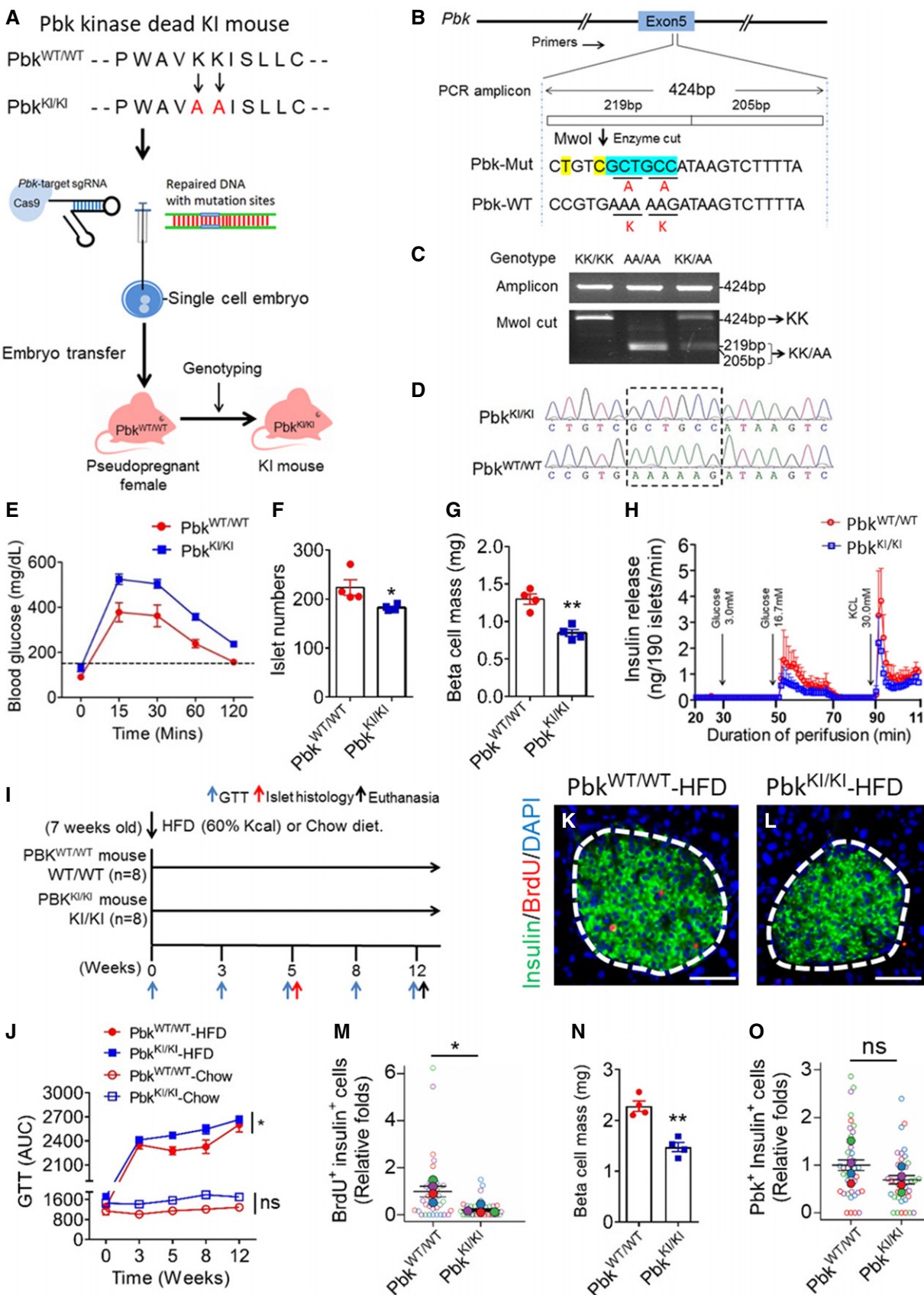

Figure 2.

Pbk mutant KI mice were born normally, matured to adulthood, and did not influence body weight of the mice (Fig EV2A and B); however, impaired glucose tolerance (IGT) was observed in male mice (Fig 2E), but not in normal female mice (data not shown). Male Pbk KI mice also had a moderate yet significant reduction (~10%) in islet number (Fig 2F) and significantly reduced beta cell mass (Fig 2G) compared to control mice at 11 weeks old. *Ex vivo* islet perfusion studies showed reduced insulin secretion in response to glucose stimulation in islets with Pbk KI, as compared with wild-type islets (Fig 2H). Together, these findings indicate a potential function of Pbk in maintenance of normal beta cell function in male mice; therefore, we used male mice for the remainder of the studies. Furthermore, these results indicate the importance of Pbk kinase activity in maintaining normal islet mass and beta cell function including glucose tolerance (GT).

## Pbk is essential for HFD-induced beta cell proliferation *in vivo*

We further investigated whether Pbk upregulation plays a crucial role in HFD-induced IGT and beta cell proliferation (Fig 2I). Male Pbk$^{KI/KI}$ or Pbk$^{WT/WT}$ mice were fed a HFD or chow diet for 12 weeks, with both groups showing similar body weight increase (Fig EV2C) and insulin tolerance (Fig EV2D). Both groups also showed a similar IGT after 3-week HFD feeding (Fig 2J; Fig EV2E). However, Pbk$^{WT/WT}$ mice showed improved IGT at 5th and 8th weeks (Fig 2J; Fig EV2F and G), indicating a compensation of beta cell function. In contrast, Pbk$^{KI/KI}$ mice showed progressive exacerbation in IGT (Fig 2J; Fig EV2F–H). Importantly, Pbk$^{KI/KI}$ mice had fewer BrdU positive beta cells (Fig 2K–M) compared to Pbk$^{WT/WT}$ mice, suggesting impairment in the compensatory proliferation. Moreover, Pbk$^{KI/KI}$ mice had reduced beta cell mass compared to Pbk$^{WT/WT}$ mice (Fig 2N), while Pbk KI did not significantly affect Pbk expression after 5 weeks of on HFD (Fig 2O), as the mutant only affected kinase activity. Taken together, we show that upregulated Pbk expression is essential for compensatory beta cell proliferation in HFD-induced obese mice and that the kinase activity of Pbk

is crucial for beta cell compensatory proliferation in response to HFD.

## JunD suppresses Pbk expression in a menin-dependent manner

To investigate how Pbk expression is upregulated in the beta cells, we genetically deleted menin using *Men1*-specific clustered regularly interspaced short palindromic repeats (CRISPR) sgRNAs (Fig 3A) and found that the menin KD upregulated Pbk expression in INS-1 cells (Fig 3A and B). Likewise, *Men1* deletion also upregulated Pbk protein and mRNA levels in PIME cells (Fig 3C and D). Together, these results demonstrated the crucial role of menin in repressing Pbk expression in beta cells. As menin interacts with JunD (Agarwal *et al*, 1999), we performed JunD KD in INS-1 cells using shRNAs (Fig 3E) and found that JunD KD led to enhanced Pbk expression at the protein and mRNA levels (Fig 3E and F). Consistently, JunD KD in PIME cells also resulted in upregulation of Pbk protein and mRNA levels (Fig 3G and H). Moreover, *Men1* deletion blocked JunD KD-induced upregulation of Pbk expression at the protein and mRNA level (Fig 3I and J). In aggregate, these findings indicate that menin is essential for JunD-mediated repression of Pbk expression in beta cells.

As JunD is a transcription factor, we searched for potential JunD binding site (JBS) surrounding the *Pbk* locus in different species using PROMO 3.0. Notably, several predicted potential JBSs were identified at the *Pbk* locus in the mouse, rat, and human genome (Fig 3K). The gene sequences of the predicted JBSs in the mouse *Pbk* promoter were listed in Appendix Table S2. Luciferase reporter assay showed that JBS2, JBS3, and JBS4-mediated JunD-induced upregulation of the reporter expression, with JBS4 yielding the highest activity (Fig 3L). To further analyze whether intact JBS4 is essential for JunD-mediated reporter gene expression, we transfected HEK293T cells with either the WT or mut-luc reporter, in combination with JunD and/or menin constructs. The reporter assay showed that WT JBS4 upregulated the reporter expression, but a point mutation in JBS4 abolished JunD-mediated induction of the

---

**Figure 3. JunD was involved in menin-mediated Pbk suppression.**

A, B   Menin knock down (KD) in INS-1 cells using a *Men1*-specific CRISPR sgRNA upregulated Pbk protein levels (A) and mRNA levels (B). qPCR data were from three independent experiments (*n* = 3). **P = 0.0093, ***P = 0.0004 (two-tailed unpaired Student's *t*-test).

C, D   *Men1* knock out (KO) in PIME cells through Cre-mediated gene editing upregulated Pbk protein level (C) and mRNA level (D). qPCR data were from three independent experiments (*n* = 3). **P = 0.0059 (two-tailed unpaired Student's *t*-test).

E, F   JunD KD in INS-1 cells using shRNA upregulated Pbk protein levels (E) and mRNA levels (F). qPCR data were from three independent experiments (*n* = 3). ***P = 0.0001 (shRNA-#1), ***P = 0.0007 (shRNA-#2) (two-tailed unpaired Student's *t*-test).

G, H   JunD KD in PIME cells using shRNA increased Pbk protein levels (G) and mRNA levels (H). qPCR data were from three independent experiments (*n* = 3). *P = 0.0236 (shRNA-#1), *P = 0.0175 (shRNA-#2) (two-tailed unpaired Student's *t*-test).

I, J   JunD KD in menin-null PIME cells moderately suppressed Pbk protein levels (I) and mRNA levels (J). qPCR data were from three independent experiments (*n* = 3). **P = 0.0030 (shRNA-#1), **P = 0.0065 (shRNA-#2) (two-tailed unpaired Student's *t*-test).

K   The predicted JunD binding sites within *Pbk* promoters of mouse, human, and rat genomes via software PROMO3.0.

L   HEK293T cells were co-transfected with a JunD-expressing plasmid and luciferase reporter constructs driven by different wild-type mouse JunD binding sites (JBSs) for 36 h, followed by luciferase assay. Three independent experiments (*n* = 3). **P = 0.0011 (site 2), **P = 0.0032 (site 3), ***P = 0.0001 (site 4) (two-tailed unpaired Student's *t*-test). ns, not statistically significant difference.

M   Luciferase reporters driven by WT or mutant JBS4 were transfected into HEK293T cells with a JunD expression plasmid or with or without menin-expressing plasmid. Luciferase assays were performed 36h after transfection. Three independent experiments (*n* = 3).

N   JunD overexpression in PIME cells suppressed Pbk protein levels (M, Lane 2 vs Lane 1), but did not affect Pbk expression in menin-null PIME cells (M, Lane 4 vs. Lane 3).

O   JunD KD suppressed PIME cell growth, but not menin-null PIME cells. Two independent experiments (*n* = 2). **P = 0.0025 (Two-way ANOVA). ns, not statistically significant difference.

Data information: Data are represented as mean ± SEM.

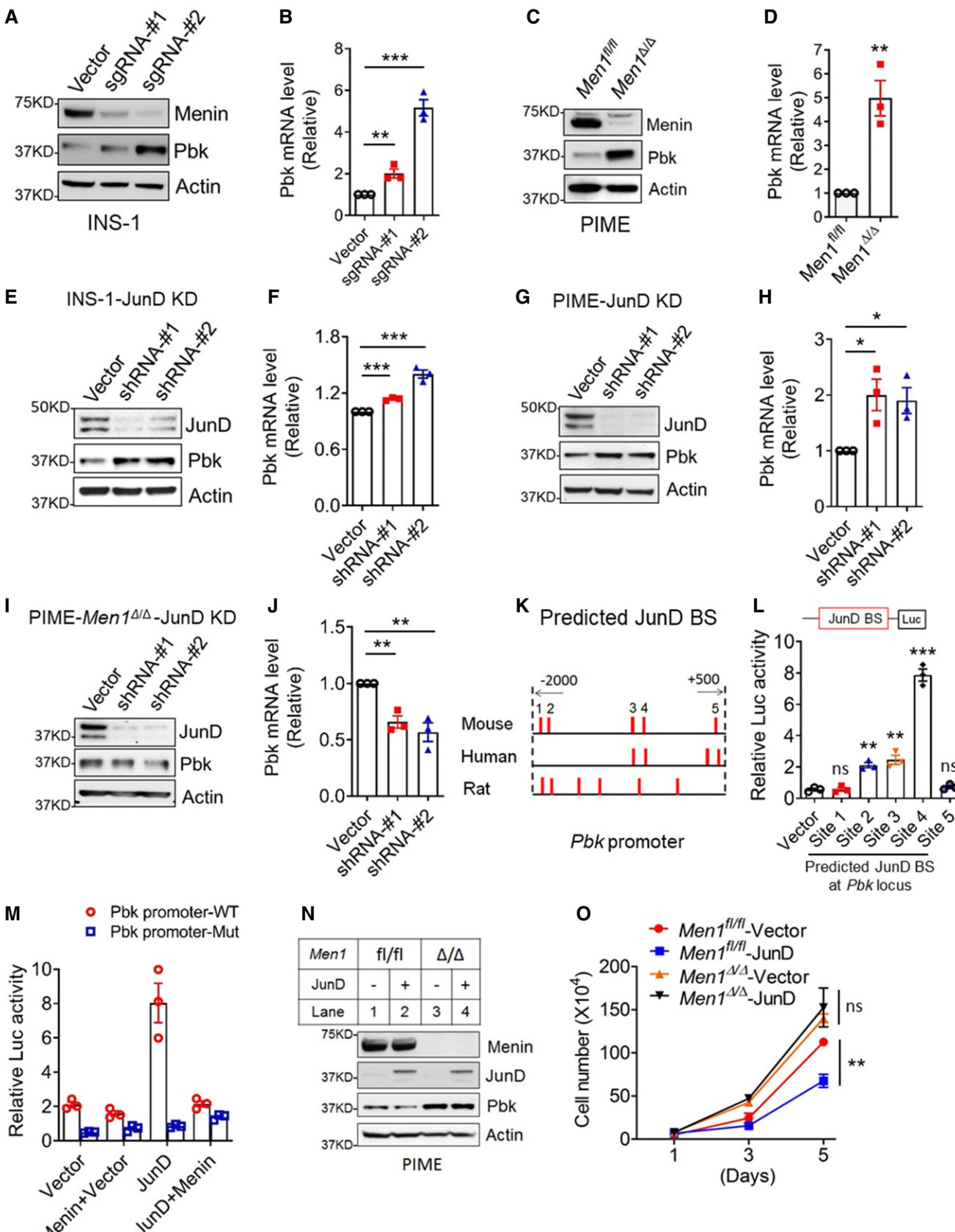

Figure 3.

reporter expression (Fig 3M), indicating the crucial role of JunD-specific binding to JBS4 in promoting the target gene transcription. Notably, JunD binding site-mediated induction of the reporter gene was completely repressed by co-transfection with the menin-expressing construct (Fig 3M), demonstrating the specific role of the JunD binding site as well as menin in mediating repression of Pbk expression.

Further supporting the above observations, ectopic expression of JunD in PIME cells modestly suppressed Pbk expression, but this effect was abolished in *Men1*-excised cells (Fig 3N). Moreover, JunD expression repressed growth of menin-expressing PIME cells, but failed to affect growth of the menin-null cells (Fig 3O). Together, these results suggest that JunD specifically binds to the *Pbk* promoter and is crucial for menin-mediated Pbk transcription suppression.

### JunD recruits menin-HDAC complex to *Pbk* promoter to suppress Pbk expression and cell proliferation

To understand how JunD and menin regulate Pbk expression, given the previous report on menin–HDAC1 interaction *in vitro* (Gobl *et al,* 1999; Kim *et al,* 2003) and our observation that menin interacted with HDAC3 and HDAC1 (Figs 4A and EV3A), we used histone deacetylase (HDACs) inhibitor vorinostat (Finnin *et al,* 1999) to treat menin-expressing PIME cells. Vorinostat upregulated Pbk expression in PIME cells in a dose-dependent manner (Fig 4B), but the upregulation was abolished by the *Men1* gene excision (Fig 4C). These results highlight the menin-dependent and HDAC-mediated repression of Pbk expression. Further, chromatin immunoprecipitation (ChIP) assays showed that JunD KD reduced JunD binding to the *Pbk* promoter, and also decreased the recruitment of menin and HDAC3, with subsequent increased histone acetylation at the *Pbk* locus in INS-1 cells (Fig 4D). Meanwhile, *Men1* KD led to reduced menin and HDAC3 binding to the Pbk promoter in INS-1 cells, but increased the detection of transcription-activating acetylated histone3 level at the *Pbk* promoter (Fig 4E and F). In addition,

menin and HDAC3, but not HDAC1, bound to the *Pbk* promoter (Figs 4D–F and EV3B). Consistently, JunD KD resulted in reduced binding of menin and HDAC3 to the *Pbk* promoter in PIME cells as well as in INS-1 cells (Fig 4G). Of note, *Men1* excision did not influence JunD binding to the *Pbk* locus (Fig 4E and F, H and I), indicating that JunD recruits menin to the Pbk promoter, but loss of menin did not affect JunD binding to the promoter.

To investigate how JunD influences menin and HDAC3 at the Pbk promoter, we sought to examine how an AP-1 family inhibitor, T5224 (Aikawa *et al,* 2008), which prevents AP-1 proteins including JunD from binding to target gene loci, affects Pbk expression and the recruitment of JunD, menin and HDAC3 at the Pbk promoter. To this end, PIME cells were treated with T5224, and ChIP assays showed that the T5224 treatment reduced the recruitment of JunD, menin, and HDAC3 to the Pbk promoter, and increased the H3 acetylation level at the promoter in two independent cell lines (Fig 4J and K). Furthermore, T5224 increased Pbk protein expression in both PIME cells (Fig 4L) and INS-1 cells (Fig EV3C). Collectively, these results demonstrate that JunD binds to the *Pbk* promotor and recruits menin and HDAC3 to repress *Pbk* transcription.

### Menin inhibitor (MI) promotes cell growth by decreasing menin binding at the *Pbk* locus through blocking the interaction of menin and JunD

We previously showed that menin binds to a short JunD peptide (Fig 5A); however, whether MIs inhibit that binding remains uncertain (Huang *et al,* 2012). We used Co-IP to show that both MI-503 and MI-463 (Borkin *et al,* 2015) inhibited the binding of menin and JunD in a dose-dependent manner, markedly inhibiting the binding at the high concentrations (Fig 5B, lane 2 vs. 5, 2 vs. 8, and bottom 2 panels). To investigate whether MI treatment of beta cells affects binding of menin, HDAC3, JunD, as well as histone acetylation of the *Pbk* promoter, we treated various beta cell lines with MI and then performed ChIP assays, showing that MI treatment reduced

**Figure 4. JunD recruits menin-HDAC3 complex to repress Pbk expression.**

A       Co-IP experiments in HEK293T cells transfected with vector and flag-HDAC3 expression plasmids. Anti-flag antibody was used to put down HDAC3 protein. WB data showed the level of menin and HDAC3 of input samples and output samples.

B, C    Different concentrations of vorinostat were used to treat PIME cells (B) and menin-null PIME cells (C) for 48 h. Pbk expression was examined using WB.

D       The control INS-1cells and JunD KD INS-1 cells were subjected to ChIP assay to detect for the binding of menin, HDAC3, JunD, and histone3 acetylation level at the *Pbk* locus. 1% input denotes the signal level normalized by the equivalent total input DNA (i.e., 1% of sonicated genome DNA) for ChIP assay. Three independent experiments ($n = 3$). *$P = 0.0430$ (JunD), *$P = 0.0480$ (Menin), ***$P = 0.0008$ (HDAC3), *$P = 0.0480$ (Ac-H3) (two-tailed unpaired Student's *t*-test). ns, not statistically significant difference.

E, F    The control INS-1cells and menin KD INS-1 cells induced by CRISPR/Cas9 system were subjected to ChIP assay to detect for the binding of menin, HDAC3, JunD, and histone3 acetylation level at the *Pbk* locus. Three independent experiments ($n = 3$). *$P = 0.0196$ (E, Menin), *$P = 0.0164$ (E, HDAC3), *$P = 0.0323$ (F, Ac-H3) (two-tailed unpaired Student's *t*-test). ns, not statistically significant difference.

G       Control and JunD KD PIME cells were subjected to ChIP assay to detect the binding of JunD, menin, HDAC3, and histone acetylation at the *Pbk* locus. Three independent experiments ($n = 3$). ***$P = 0.0007$ (JunD), *$P = 0.0486$ (Menin), *$P = 0.0117$ (HDAC3), **$P = 0.0012$ (Ac-H3) (two-tailed unpaired Student's *t*-test). ns, not statistically significant difference.

H, I    PIME cells and menin-null PIME cells were subjected to ChIP assay to detect the binding of menin, HDAC3, JunD (H) as well as histone acetylation level (I) at the *Pbk* locus. Three independent experiments ($n = 3$). *$P = 0.0226$ (H, Menin), **$P = 0.0084$ (H, HDAC3), **$P = 0.0041$ (I, Ac-H3) (two-tailed unpaired Student's *t*-test). ns, not statistically significant difference.

J, K    The effect of T5224 on the binding of JunD, menin, HDAC3, and acetylated histone3 at the *Pbk* locus in PIME cells (J) or INS-1 cells (K). Three independent experiments ($n = 3$). *$P = 0.0490$ (J, JunD), *$P = 0.0479$ (J, Menin), *$P = 0.0154$ (J, HDAC3), **$P = 0.0029$ (J, Ac-H3), *$P = 0.0104$ (K, JunD), **$P = 0.0024$ (K, Menin), ***$P = 0.0002$ (K, HDAC3), ***$P = 0.0001$ (K, Ac-H3), (two-tailed unpaired Student's *t*-test). ns, not statistically significant difference.

L       PIME cells were treated with various doses of T5224 for 48 h, followed by detecting Pbk expression using WB.

Data information: Data are represented as mean $\pm$ SEM.

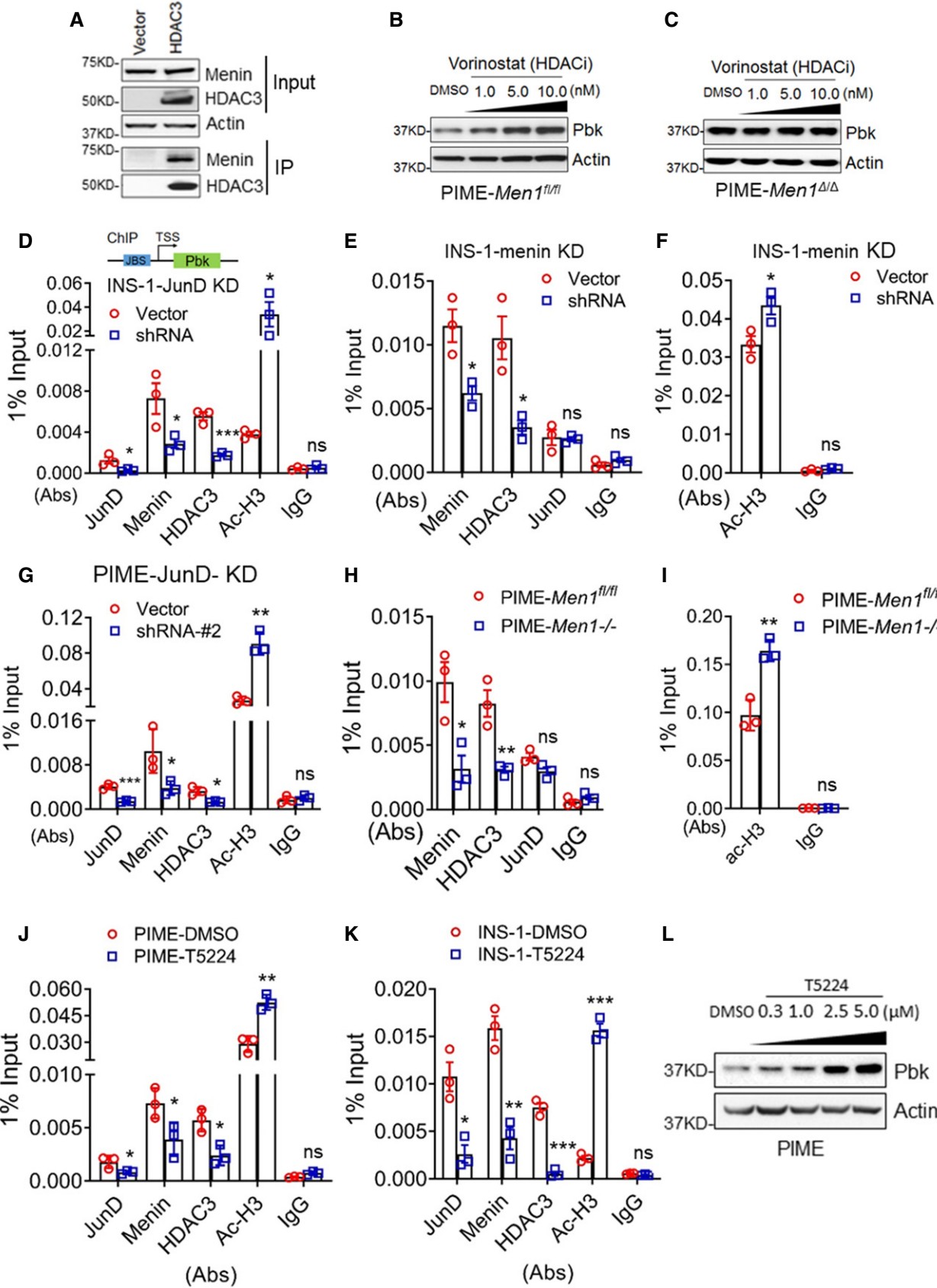

Figure 4.

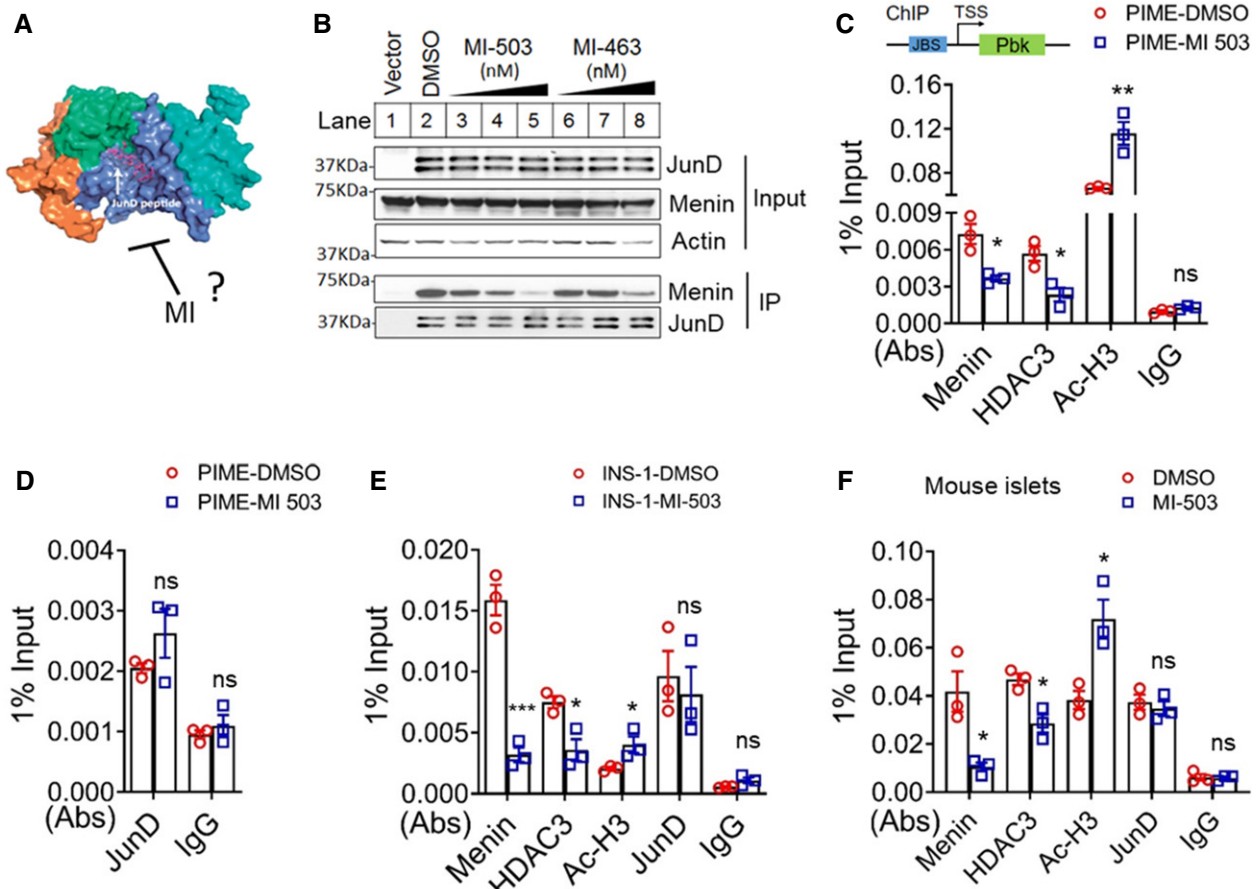

**Figure 5. Pharmacological inhibition of menin decreases the binding of menin at the *Pbk* locus.**

A   Crystal structure of the menin–JunD$_{MBM}$ complex (Adapted from Huang *et al*, 2012). MI potentially blocks the interaction of menin and JunD.

B   Co-IP in HEK293T cells transfected with vector or a flag-JunD-expressing plasmid upon treatment with DMSO, MI-503 (30, 100, 300 nM from lane 3 to lane 5, respectively), or MI-463 (30, 100, 300 nM from lane 6 to lane 8, respectively).

C, D   ChIP assay to detect the binding of menin, HDAC3, JunD, and histone3 acetylation at the *Pbk* locus in vehicle or MI-503-treated PIME cells. Three independent experiments (*n* = 3). *P* = 0.0132 (Menin), *P* = 0.0153 (HDAC3), **P* = 0.0092 (Ac-H3) (two-tailed unpaired Student's *t*-test). ns, not statistically significant difference.

E   ChIP assay to detect the binding of menin, HDAC3, JunD, and histone3 acetylation at the *Pbk* locus in vehicle or MI-503-treated INS-1 cells. Three independent experiments (*n* = 3). ***P* = 0.0008 (Menin), **P* = 0.0183 (HDAC3), **P* = 0.0485 (Ac-H3) (two-tailed unpaired Student's *t*-test). ns, not statistically significant difference.

F   Mouse primary islets treated with MI-503 for 3 days were subjected ChIP assay to detect the binding of menin, HDAC3, JunD, as well as histone acetylation level at the *Pbk* locus. Three independent experiments (*n* = 3). **P* = 0.0218 (Menin), **P* = 0.0145 (HDAC3), **P* = 0.0184 (Ac-H3) (two-tailed unpaired Student's *t*-test). ns, not statistically significant difference.

Data information: Data are represented as mean ± SEM.

binding of menin and HDAC3 to the *Pbk* promoter, but increased histone acetylation at the promoter in PIME cells (Fig 5C). In contrast, the MI treatment did not affect JunD binding to the promoter (Fig 5D). Consistently, MI treatment of INS-1 cells also reduced binding of menin and HDAC3, but not JunD, and increased histone acetylation of the promoter (Fig 5E). Furthermore, to evaluate the relevance of these results to primary islet cells, we isolated primary pancreatic islets from normal C57BL/6 mice and treated the primary islets with vehicle and MI. The ChIP assay showed that MI reduced binding of menin and HDAC3, but not JunD, to the *Pbk* promoter, but increased histone 3 acetylation at the promoter (Fig 5F). Together, these results indicate that MI blocks the menin–JunD

interaction, resulting in reduced binding of menin and HDAC3 to the *Pbk* promoter, leading to increased histone 3 acetylation and upregulation of Pbk transcription.

To further determine whether MI-induced Pbk expression is menin dependent, we used MI-463 to treat menin-expressing PIME cells and found that MI treatment promoted Pbk expression in a dose-dependent manner (Fig 6A and B). In contrast, MI-induced Pbk upregulation was not observed in PIME cells with *Men1* deletion (Fig 6C and D). Similarly, treatment with MI-503, an analog of MI-463, also upregulated Pbk expression at the protein and mRNA level in PIME cells in a menin-dependent manner (Fig EV4A–D). Moreover, treating PIME cells or INS-1 cells with MI increased cell

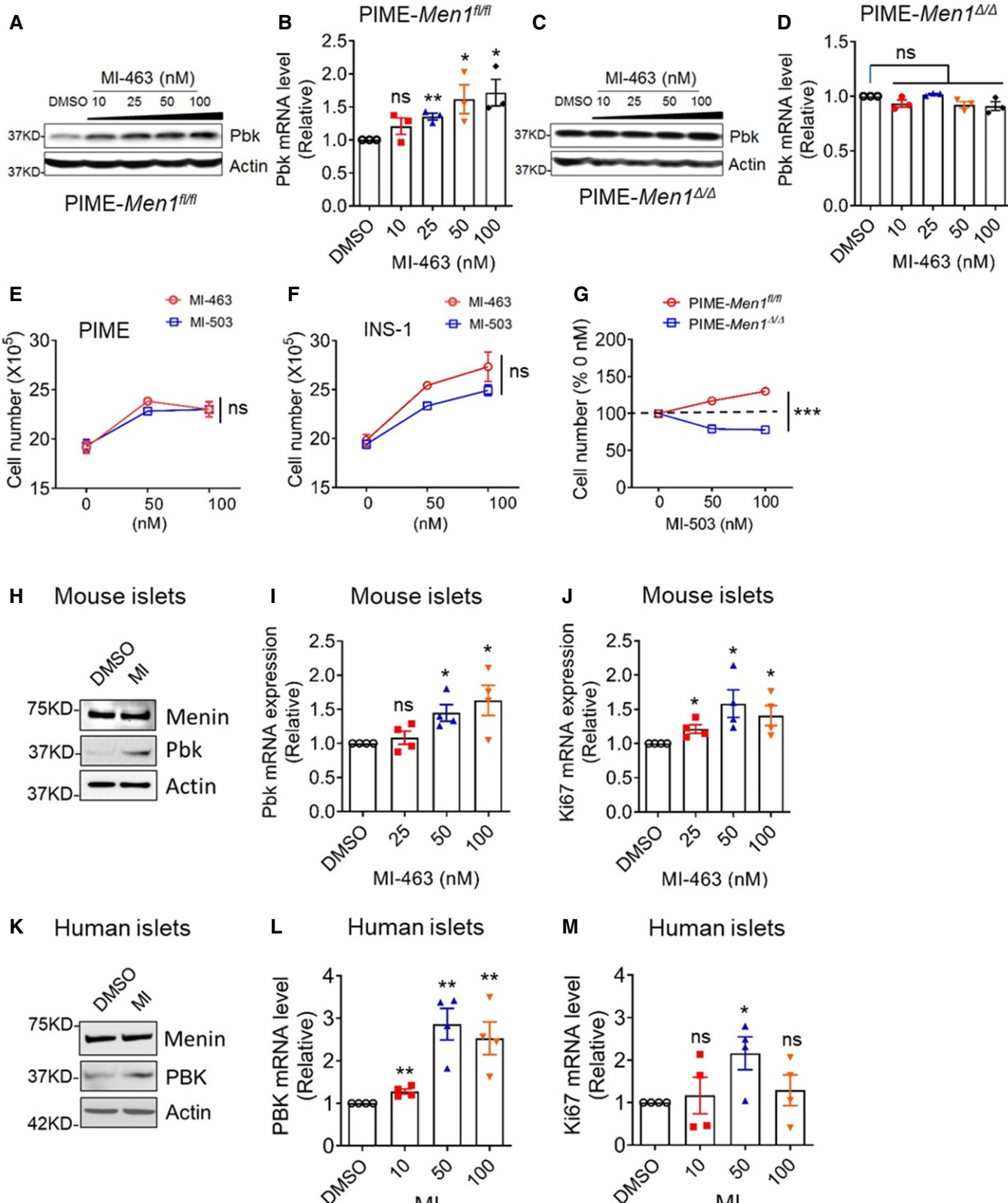

**Figure 6.**

**Figure 6. Menin inhibitor (MI) increased Pbk expression and correlates with beta cell proliferation *in vitro*.**

A, B    PIME cells were treated with various doses of MI-463 for 48 h, followed by detecting Pbk expression using WB (A) and qPCR (B). qPCR data were from three independent experiments ($n = 3$). **$P = 0.0022$ (25 nM), *$P = 0.0479$ (50 nM), *$P = 0.0233$ (100 nM) (two-tailed unpaired Student's $t$-test). ns, not statistically significant difference.

C, D    Menin-null PIME cells were treated with various concentrations of MI-463 for 48 h, followed by detecting Pbk expression with WB (C) and qPCR (D). qPCR data were from three independent experiments ($n = 3$). ns, not statistically significant difference (two-tailed unpaired Student's $t$-test).

E, F    MI-503 and MI-463 treatment promotes growth of PIME cells (E) and INS-1 cells (F). Three independent experiments ($n = 3$). ns, not statistically significant difference (Two-way ANOVA).

G       MI-503 increases proliferation of menin-expressing PIME cells but not menin-null PIME cells. Three independent experiments ($n = 3$). ***$P < 0.0001$ (Two-way ANOVA).

H–J     5 days of MI-463 treatment increases the expression of Pbk at the protein level (H) and mRNA level (I), as well as the proliferation marker ki67 mRNA levels (J) in mouse primary islets. qPCR data were from three independent experiments ($n = 3$). *$P = 0.0105$ (I, 50 nM), *$P = 0.0287$ (I, 100 nM), *$P = 0.0150$ (J, 25 nM), *$P = 0.0273$ (J, 50 nM), *$P = 0.0314$ (J, 100 nM) (two-tailed unpaired Student's $t$-test). ns, not statistically significant difference.

K–M     5 days of MI-503 treatment impacts the expression of Pbk at protein level (K) and mRNA level (L), as well as ki67 mRNA levels (M) in human primary islets (donor number is four). qPCR data were from three independent experiments ($n = 3$). **$P = 0.0036$ (L, 10 nM), **$P = 0.0024$ (L, 50 nM), **$P = 0.0073$ (L, 100 nM), *$P = 0.0237$ (M, 50 nM) (two-tailed unpaired Student's $t$-test). ns, not statistically significant difference.

Data information: Data are represented as mean ± SEM.

growth by 20-30% *in vitro* (Fig 6E and F). MI treatment increased cell growth of menin-expressing PIME cells but not menin-null cells (Fig 6G). These results demonstrate that MI-induced beta cell proliferation depends on the expression of the target protein menin, highlighting the specificity of this effect.

We next examined the effects of MI on primary human and mouse islets *ex vivo*. MI treatment upregulated Pbk expression at the protein and mRNA levels in a dose-dependent manner in mouse primary islets (Fig 6H and I), consistent with the increase in Ki67 mRNA level in mouse islets with MI treatment (Fig 6J). Similar results were also observed in human primary islets (Fig 6K–M). Collectively, these results demonstrate that MI upregulates Pbk expression via blocking the JunD–menin interaction, influencing histone acetylation marks, and promoting beta cell proliferation in a menin-dependent manner.

**Pbk is essential for MI-induced beta cell proliferation *in vivo***

To assess the *in vivo* effect of MI and Pbk on beta cell proliferation, Pbk^KI/KI or Pbk^WT/WT male mice (8–12 weeks old) were treated with MI-463 for 8 weeks (Fig 7A), with no difference in body weight observed between the two groups on a standard chow diet (Appendix Fig S3A and B). As expected, Pbk^KI/KI mice showed impaired IGT as compared to Pbk^WT/WT mice, with significantly increased area under the curve (AUC) (Fig 7B and C). Treating Pbk^WT/WT mice with MI-463 improved GT, whereas Pbk^KI/KI showed no improvement in GT with MI-463 treatment (Fig 7B and C). Moreover, MI-463 significantly enhanced GSIS in Pbk^WT/WT mice, but there was no enhancement in Pbk^KI/KI mice (Fig 7D). Additionally, immunofluorescence staining (Fig 7E-L) showed that MI treatment increased the number of beta cells with the proliferation marker Ki67 in Pbk^WT/WT mice (Fig 7F vs. 7E, 7 M), but not in Pbk^KI/KI mice (Fig 7H vs. 7G, 7 M). Consistently, IF staining showed that MI treatment of wild-type mice increased Pbk expression in beta cells (Fig 7J vs. 7I, 7N). Of note, MI treatment also increased Pbk expression in Pbk^KI/KI mice, as Pbk KI has the same promoter as Pbk WT and was therefore still regulated by menin (Fig 7L vs. 7K, and 7N). However, Pbk KI completely lost the ability to sustain the MI-induced beta cell proliferation (Fig 7H and 7M). Further, MI treatment also increased the beta cell mass in Pbk^WT/WT mice, but Pbk KI abolished MI treatment-mediated increases in beta cell mass (Fig 7O). In aggregate, these results demonstrate that Pbk kinase activity plays a crucial role in MI-induced beta cell proliferation in mice on a standard chow diet.

**Figure 7. Pbk is essential for MI-increased beta cell proliferation and improved GT *in vivo*.**

A       A diagram of experimental design. 8- to 12-week-old Pbk^KI/KI and Pbk^WT/WT mice were treated with MI-463 at 70 mg/kg ($n = 4$) or vehicle ($n = 4$) by gavage daily.

B, C    GTT data from Pbk^KI/KI and Pbk^WT/WT mice ($n = 4$ for each group) after 8 weeks of MI-463 or vehicle treatment (B) (***$P < 0.0001$ (Two-way ANOVA), ns, not statistically significant difference) and AUC of the GTT results (C) (***$P = 0.0005$ (two-tailed unpaired Student's $t$-test), ns, not statistically significant difference). The mice were starved overnight prior to the test (16 h). Dot line, fasting blood glucose number > 150 was designated as hyperglycemia.

D       *In vivo* GSIS test on 11-week-old male Pbk^KI/KI and Pbk^WT/WT mice ($n = 4$ for each group). The mice were starved overnight prior to the test (16 h). Y-axis represents the ratio of insulin level at 15 min over insulin level at 0 min with glucose challenge. **$P = 0.0042$ (two-tailed unpaired Student's $t$-test). ns, not statistically significant difference.

E–L     Representative images of insulin (green) and Ki67 (red) double staining on mouse islets with 8 weeks MI treatment (E–H), and insulin (green) or Pbk (red) double staining on mouse islets (I–L). Nuclei were labeled by DAPI (blue). Islet area was circled by white dashed line. Ki67 and Pbk staining are denoted by white arrows. Scale bars: 50 μm.

M, N    Quantification of the percentages of Ki67 staining-positive beta cells (M) and Pbk staining-positive beta cells (N). Four mice for each group, and 5–10 islet images per mouse were analyzed. **$P = 0.0026$ (M, WT/WT-Mock : WT/WT-MI-463), **$P = 0.0013$ (M, KI/KI-Mock : KI/KI-MI-463), **$P = 0.0011$ (N, WT/WT-Mock : WT/WT-MI-463), ***$P = 0.0009$ (N, KI/KI-Mock : KI/KI-MI-463) (two-tailed unpaired Student's $t$-test). ns, not statistically significant difference.

O       Beta cell mass comparison between MI-463- and vehicle-treated Pbk^KI/KI mice and Pbk^WT/WT mice ($n = 4$ for each group). *$P = 0.0174$ (two-tailed unpaired Student's $t$-test). ns, not statistically significant difference.

Data information: Data are represented as mean ± SEM.

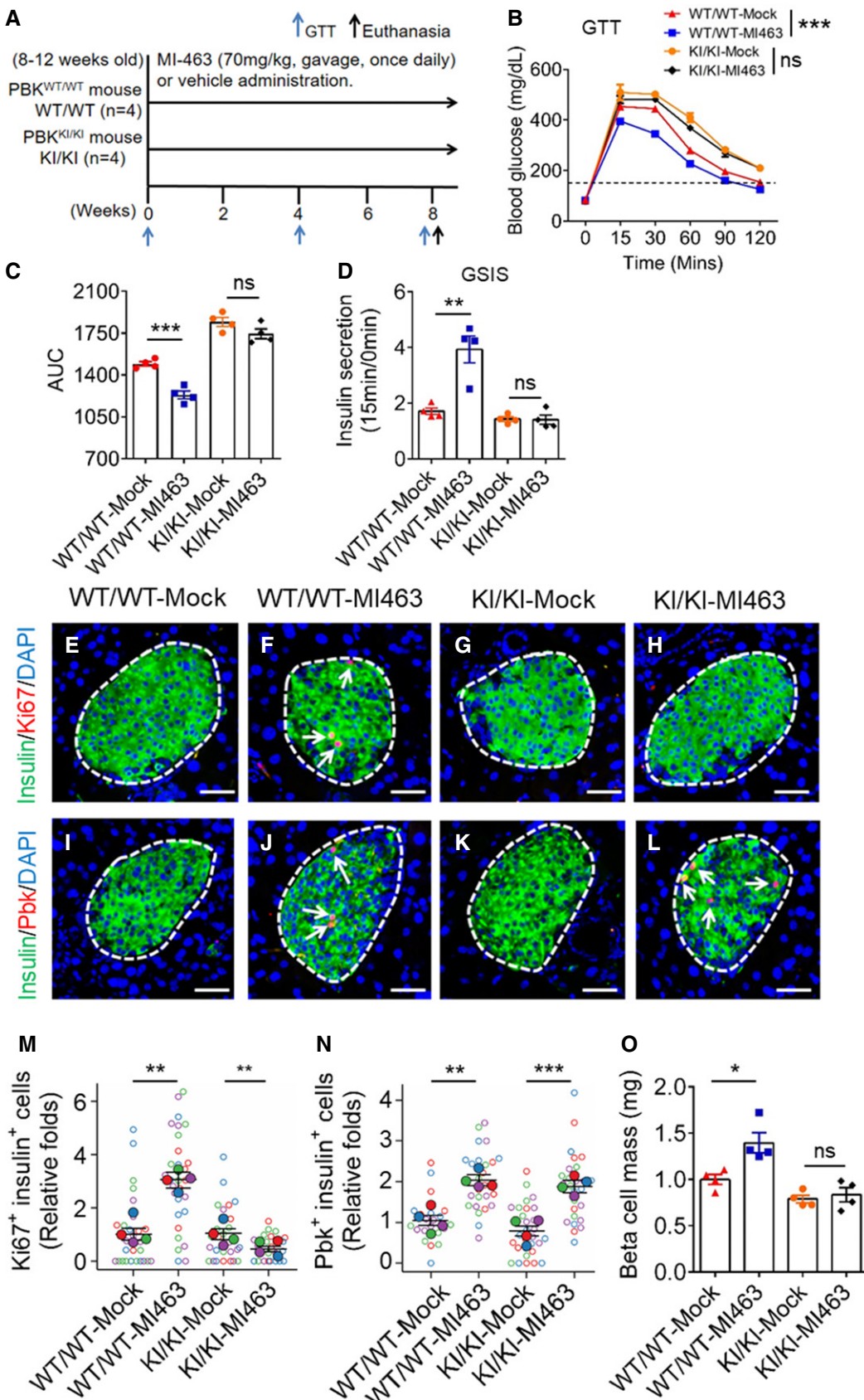

**Figure 7.**

## Pbk is essential for MI-induced improvement of hyperglycemia and IGT in HFD-induced diabetic mice

To investigate the potential effect of MI on HFD-induced diabetes, hyperglycemic HFD mice were treated with either MI-463 or vehicle (Fig 8A). Notably, MI treatment decreased the fasting glucose level in mice, as compared with vehicle (Figs 8B and EV5A and B). Prior to MI treatment, HFD-fed mice showed IGT (Figs 8C and EV5C), as expected. In control mice, the IGT remained unchanged throughout the 12 weeks observed (Figs 8C and EV5A and 5C–G). In contrast, there was consistent improvement in IGT in MI-treated mice starting at the 6th week that persisted through the 12th week (Figs 8C and EV5E–G). Collectively, these results demonstrate that MI treatment significantly reduced hyperglycemia and improved IGT in HFD-induced diabetic mice.

During MI administration to mice, body weight showed a slight fluctuation that was not statistically significant (Fig EV5H). Insulin tolerance test (ITT) indicated that the insulin sensitivity was moderately improved in mice with MI treatment (Fig EV5I), and this might partly contribute to the enhanced GT profile of the mice. However, MI treatment did not change food consumption (Fig EV5J), but rather improved IGT (Fig 8C; Fig EV5C–G). The MI treatment did not significantly change adiposity index (Fig EV5K), prompting us to further investigate the impact of the MI treatment on beta cell mass in pancreatic islets. The histological studies indicate that MI treatment indeed increased the number of BrdU and insulin co-staining cells (Fig 8D–F), as well as the insulin-stained area of pancreas sections (Fig 8G). IF staining also showed that MI treatment increased the number of Pbk-expressing beta cells in the islets of HFD-fed mice (Fig 8H–J), with 65% of Pbk-positive cells also being BrdU positive, indicating a Pbk-driven cell proliferation (Fig 8K).

To further define the role of Pbk in MI-mediated improvement of HFD-induced diabetes, we investigated the impact of MI treatment on GT on $Pbk^{KI/KI}$ and $Pbk^{WT/WT}$ HFD mice (Fig 8L). Interestingly, we found that a 3-week feeding of HFD led to similar IGT in both

$Pbk^{KI/KI}$ and $Pbk^{WT/WT}$ mice (Fig 8M and N), indicating that Pbk is not crucial for regulating the acute development of HFD-induced IGT. However, the treatment of HFD $Pbk^{WT/WT}$ mice with MI-463 for 8 weeks improved IGT (Fig 8O and P). Notably, Pbk KI completely abolished the MI treatment-induced improvement in IGT (Fig 8O and P), but did not affect bodyweight of either strain (Fig EV5L and M). In aggregate, these findings provide strong evidence that MI augments Pbk expression and beta cell proliferation in diabetic mouse islets, and that Pbk kinase activity is essential for MI-induced beta cell proliferation and increased mass, and improvement of IGT in diet-induced diabetic mouse models.

# Discussion

Targeting genetic pathways to enhance proliferation of endogenous pancreatic beta cells may serve as a promising approach to improve diabetes therapy. In the current study, we demonstrated the essential role of Pbk in promoting compensatory beta cell proliferation. Pbk is normally expressed in several select tissues (Abe et al, 2000; Zhu et al, 2007) and a subset of cancers including breast cancer, ovarian cancer, and leukemia (Park et al, 2006; Ikeda et al, 2016; Ishikawa et al, 2018), but is mainly restricted to non-dividing tissues (Gaudet et al, 2000; Zhao et al, 2001). Herein, we uncover for the first time the essential role of Pbk in regulating beta cell function under HFD-induced stress situation. While the interaction of menin recruits HDAC complex to repress transcription of reporter genes in transformed cells (Gobl et al, 1999; Kim et al, 2003), our studies are the first to demonstrate menin directly represses transcription of endogenous Pbk through interacting with both JunD and HDAC3, to repress compensatory beta cell proliferation in obese conditions.

We elucidated the underlying mechanisms whereby menin interacts with JunD and HDAC3, which then reduces histone H3 acetylation, a marker for active chromatin, leading to epigenetic repression of Pbk. According to this working model, menin/JunD-mediated

---

**Figure 8. The crucial role of Pbk in MI-mediated hyperglycemia reversion and the amelioration of IGT in HFD-induced diabetic mice.**

A    A diagram for treating mice with MI. HFD-fed mice (18 weeks old) were treated with MI-463 at 70 mg/kg (n = 5) or vehicle (corn oil, n = 5) by gavage daily.

B    Fasting blood glucose levels of MI-463- or vehicle-treated mice were measured at the indicated times during administration. n = 5 for each group of mice. Dot line, fasting blood glucose number > 150 was designated as hyperglycemia. **P = 0.0026 (Two-way ANOVA).

C    AUC comparison of IPGTT data from MI-463- or vehicle-treated mice during 12 weeks administration. n = 5 for each group of mice. **P = 0.0024 (Two-way ANOVA).

D, E    Double staining for insulin (green) and BrdU (red) of mice islets after 12 weeks of vehicle (D) or MI (E) administration. Islet area was circled by white dashed line. Nuclei were labeled by DAPI (blue). BrdU staining is indicated by arrows. Scale bar: 50 μm.

F, G    Quantification of BrdU-positive beta cells (F) and beta cell mass (G) of MI-463-treated HFD-fed mice or control mice. Five mice for each group, and 5–10 islet images per mouse were analyzed. **P = 0.0059, *P = 0.0103 (two-tailed unpaired Student's t-test).

H, I    Double staining for insulin (green) and Pbk (red) of mice islets after 12 weeks of vehicle (H) or MI (I) administration. Islet area was circled by white dashed line. Nuclei were labeled by DAPI (blue). Pbk staining is indicated by arrows. Scale bar: 50 μm.

J    Quantification of Pbk staining-positive β-cells. Five mice for each group, and 10 islet images per mouse were analyzed. *P = 0.0120 (two-tailed unpaired Student's t-test).

K    Quantification of the percentage with pbk and BrdU co-staining cells among total Pbk positive cells in HFD-fed mouse islets. The mean values of five mice from 5 sections for each mouse were presented.

L    MI-463 treatment schedule of $Pbk^{KI/KI}$ and $Pbk^{WT/WT}$ mice with HFD feeding. Seven-week-old male $Pbk^{KI/KI}$ and $Pbk^{WT/WT}$ mice were fed with HFD (n = 6 for each group). Four weeks later, HFD-fed $Pbk^{KI/KI}$ and $Pbk^{WT/WT}$ mice were started to treat with MI-463 or vehicle (i.e., n = 3 for each group) for 8 weeks.

M, N    GTT data of $Pbk^{KI/KI}$ and $Pbk^{WT/WT}$ mice (n = 6 for each group) with 3 weeks HFD feeding (M) (***P < 0.0001 (Two-way ANOVA)) and AUC comparison of GTT results (N) (***P < 0.0001 (WT/WT-Chow : WT/WT-HFD), ***P = 0.0007 (KI/KI-Chow : KI/KI-HFD) (two-tailed unpaired Student's t-test)).

O, P    GTT data for MI-463- or vehicle-treated $Pbk^{KI/KI}$ or $Pbk^{WT/WT}$ mice (n = 3 for each group) on 8-week HFD (O) (*P = 0.0133 (Two-way ANOVA), ns, not statistically significant difference) and AUC of GTT data (P) (*P = 0.0442 (two-tailed unpaired Student's t-test), ns, not statistically significant difference).

Data information: Data are represented as mean ± SEM.

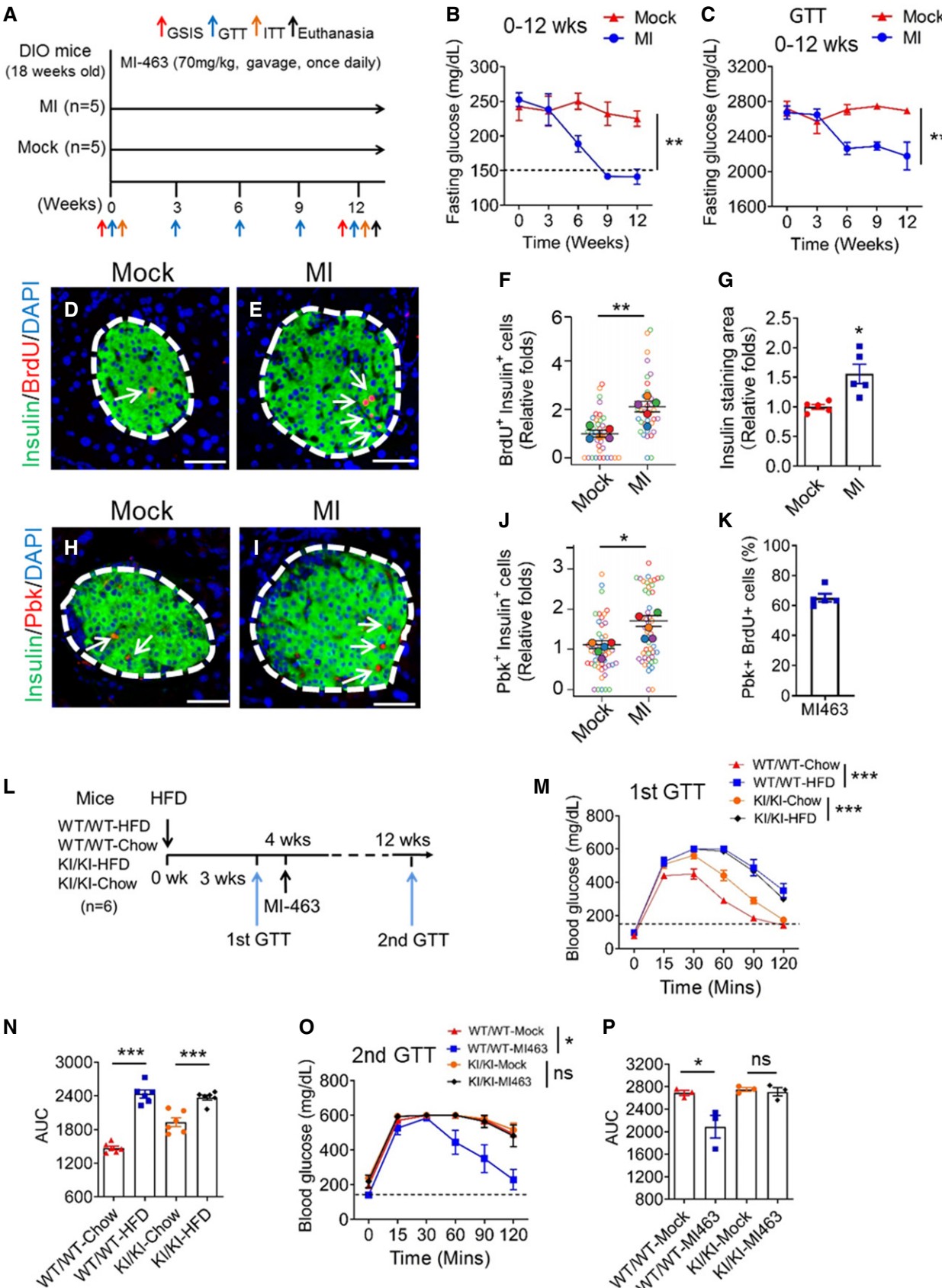

Figure 8.

suppression of Pbk expression contributes to the low level of beta cell proliferation under normal conditions. In contrast, with a HFD Pbk is upregulated, leading to increased compensatory beta cell proliferation. These findings are consistent with our and others' previous observation of upregulated Pbk expression in human and mouse pancreatic islets with loss-of-function *MEN1/Men1* mutations. Importantly, Pbk is essential for HFD-induced beta cell compensatory proliferation, as Pbk$^{KI/KI}$ mice failed to invoke compensatory beta cell proliferation when challenged with HFD (Fig 2K–N). Most importantly, blocking the menin interaction by MI derepresses Pbk expression and increases beta cell proliferation, indicating the repressive role of the menin/JunD/ Pbk axis in regulating compensatory beta cells. Other possible mechanisms are certainly possible that warrant further investigation. It was reported that transcription factor FoxM1 is required for HFD-induced beta cell proliferation in mice (Golson *et al*, 2010); therefore, it is possible that HFD may influence FoxM1 to modulate Pbk expression. On the other hand, it is also conceivable that HFD indirectly affects the recruitment of JunD/menin/HDAC3 to the *Pbk* promoter.

While Pbk/TOPK was reported to phosphorylate several protein substrates including p53-related protein kinase (PRPK) in colorectal cancer cells (Zykova *et al*, 2017) and Cdk1/cyclin B1/PRC1 kinase substrate complex (Abe *et al*, 2007), our preliminary *in vitro* studies did not detect obvious phosphorylation of menin or JunD by Pbk using the anti-menin S487 phosphorylation antibody or anti-JunD S100 antibody, respectively (data not shown). Rather, we found the purified Pbk as well as the *Men1* deletion-induced Pbk from cell lysates phosphorylated Erk2 in the *in vitro* kinase assay. However, the correlation between Pbk-enhanced Erk1/2 phosphorylation and beta cell proliferation needs further investigation.

While JunD was reported to interact with menin to regulate gene transcription (Agarwal *et al*, 1999), it was not previously known that JunD directs menin to regulate a key endogenous menin target, especially *in vivo*. Here, we found that JunD recruits menin and HDAC3 in beta cells to repress expression of Pbk, indicating that the menin/JunD/Pbk axis that is crucial for regulation of beta cell proliferation. JunD was previously linked to regulating oxidative stress in fibroblasts and endothelial cells (Gerald *et al*, 2004; Paneni *et al*, 2013), and upregulated in beta cells during metabolic stress like T2D, contributing to redox imbalance (Good *et al*, 2019). JunD$^{-/-}$ mice have a shortened lifespan and develop hyperinsulinemic hypoglycemia (Laurent *et al*, 2008), but the underlying mechanism was not clear. Our findings, coupled with previous knowledge of the involvement of JunD in beta cell biology, highlight the new role and mechanism for JunD to link menin and Pbk in maintaining beta cell function.

To further understand the role of menin/JunD/Pbk axis in maintaining beta cell homeostasis, we established a Pbk kinase inactive (KI) mouse model. Using this KI mouse model, we demonstrated that Pbk is crucial for normal beta cell proliferation and maintenance of normal GT and GSIS, as well as MI-induced beta cell proliferation and amelioration of diabetes in HFD mice. Of note, even though at a later stage, both Pbk$^{WT/WT}$ and Pbk$^{KI/KI}$ mice developed impaired GT (Fig 8O and P), Pbk function is still essential for MI-induced amelioration of IGT. These data highlight the crucial role of Pbk as a downstream pro-proliferative target normally repressed by menin but derepressed (upregulated) by MI. Nevertheless, these results do not rule out other targets regulated by MIs that may also

play a role in ameliorating HFD-induced diabetes. In our mouse model, we also found that male mice with Pbk$^{KI/KI}$ selectively developed IGT, whereas female mice did not (data not shown). As elucidated by previous studies, estrogen is pro-proliferative for beta cells (Choi *et al*, 2005; Le May *et al*, 2006), protects beta cells from apoptosis through estrogen receptor alpha, and prevents type I diabetes in mice (Le May *et al*, 2006). Thus, the select effect of Pbk kinase dysfunction KI on male mice may be attributed to lower levels of estrogen compared to female mice.

Although we showed that Pbk kinase activity is crucial for MI-mediated or HFD-induced beta cell proliferation, previous studies reported that newly generated beta cells are less functional in producing or secreting insulin in response to glucose stimulation (Russ *et al*, 2008; Scharfmann *et al*, 2014). Even though our studies were not designed to discern whether the newly generated beta cells are less functional, our findings demonstrate that MI-induced increases in islet mass are correlated with increased insulin secretion (Fig 7D and O). Moreover, glucose-stimulated insulin secretion was decreased in islets of Pbk$^{KI/KI}$ mice *ex vivo* (Fig 2H), suggesting that Pbk is crucial for MI treatment-enhanced GSIS in mice. Importantly, Pbk KI blocked the MI-induced increase in islet mass as well as insulin secretion, suggesting that the MI-increased islet mass at least partially contributes to increased insulin secretion. As for the role of Pbk kinase in regulating insulin secretion, as ectopic expression of Pbk in INS-1 cells increased phosphorylated Erk1/2 level (Fig EV1B), coupled with previous reports that activated Erk1/2 is involved in glucose-mediated remodeling of F-actin leading to insulin secretion from beta cells (Kalwat & Thurmond, 2013; Yang *et al*, 2016), it is possible that Pbk is also partly involved in facilitating insulin secretion by regulating Erk1/2. Nevertheless, further work is necessary for elucidating the link between Pbk and insulin secretion.

Gene expression databases show a limited tissue expression profile of Pbk, confined mainly to reproductive organs in adult humans or mice (Appendix Fig S1A and B) (Abe *et al*, 2000; Gaudet *et al*, 2000), and the upregulation of Pbk expression was only observed in islets but not in other metabolically active organs such as liver or muscle of a HFD mice (Fig 1B). Moreover, Pbk KI in mice did not influence body weight (Fig EV2A and B). Considered together, Pbk KI mainly influences islets and contributes to the diabetes-related phenotypes at least partly through regulating the islet function.

Our data also showed that Pbk upregulation in a limited number of beta cells is crucial for HFD or MI treatment-induced beta cell proliferation, as Pbk KI-blocked HFD or MI treatment-induced beta cell proliferations. Moreover, upregulation of Pbk only in a limited number of beta cells could be a safe way to drive modest cell proliferation, while not increasing beta cell tumorigenesis.

Unlike other commonly used drugs such as metformin or GLP-1, which acutely improve glucose control in diabetes, MI treatment epigenetically regulates the menin/JunD/Pbk axis, therefore taking a relatively longer time to increase beta cell mass and improve glucose control in diabetes. Thus, MI may represent a new class of drugs that potentially improves the overall function of islets with a sustained effect over a long period of time.

In summary, we found that the menin/JunD/Pbk axis normally acts as a suppressive mechanism that controls pancreatic beta cell proliferation and mass, regulating the development of T2D and response to MI treatment. Importantly, Pbk is essential for the HFD-induced compensatory beta cell proliferation and is directly

regulated by the menin/JunD axis and HDAC-mediated histone acetylation. Use of MI to block the interaction between menin and JunD induces Pbk expression, increases beta cell proliferation, and ameliorates HFD-induced diabetes. These findings not only unravel a novel menin/JunD/Pbk axis for regulating compensatory proliferation of beta cells, but also highlight the menin/JunD/Pbk axis as a novel target for developing more effective therapy for T2D with potential long lasting effects.

# Materials and Methods

### Mice husbandry

All mouse experiments implemented in this study followed NIH Guide for the Care and Use of Laboratory Animals and also according to the IACUC standards following ethics approval by the animal committee at the University of Pennsylvania (UPenn). 6-week-old C57BL/6 male mice, 14-week-old HFD-induced obese (DIO) mice and their male lean littermate controls, and 18-week-old DIO mice were purchased from Jackson Labs. All the mice were housed in a specific pathogen-free facility with a 12 h light-dark cycle at room temperature.

### Pbk kinase dysfunctional mutant knock-in (KI) (Pbk$^{KI/KI}$) mice

A CRISPR-based approach was utilized for generation of Pbk kinase dysfunction mutant knock-in (KI) mice (Pbk$^{KI/KI}$ mice) model (JAX Stock No. 036225) in this study. Cas9-Pbk target sgRNA and ssDNA repair template with K$^{64}$K$^{65}$ to AA mutation on Pbk were microinjected into fertilized eggs. The eggs were implanted into foster female mice. The pups from the female mice were genotyped to determine the KI efficiency. The tail DNA was extracted and amplified by specific primers. The PCR products were performed RFLP analysis and sequenced to confirm the presence of mutant gene sites. Homozygous Pbk mutant KI mice and their non-mutated litter mates, utilized as control mice, were used.

### Culture of cell lines

INS-1 cells, a rat insulinoma cell line, were cultured in RPMI 1640 media supplemented with 2 mM L-glutamine, 1 mM sodium pyruvate, 10% (v/v) FBS, 10 mM HEPES, 100 units/ml penicillin, 100 μg/ml streptomycin, and 50 μM β-mercaptoethanol. Cells were cultured in a humidified incubator at 37°C with 5% $CO_2$.

Mouse Pancreatic Islet-derived *Men1* Excisable cells (PIME cells) were isolated from a *Men1*$^{fl/fl}$; Cre-ER mice and cultured in Dulbecco's Modified Eagle's Medium (DMEM) with 10% (v/v) FBS, 100 units/ml penicillin, and 100 μg/ml streptomycin at 37°C with 5% $CO_2$. PIME cells show higher expression profile of pancreatic progenitor-related genes, such as Sox9, Hnf1b, and Onecut1 (Hnf6) than a beta cell line, beta HC9 (Appendix Fig S1A and B), but also own similar beta cell-related gene expression profiles (Appendix Fig S1C). In addition, they have a *Men1* gene-excisable characteristic (Appendix Fig S1D-F).

HEK293T cells were purchased from American Type Culture Collection (ATCC) and cultured in DMEM with 10% (v/v) FBS, 100 units/ml penicillin and 100 μg/ml streptomycin, at 37°C with 5% $CO_2$.

### Isolation and culture of mouse and human islets

Islets from DIO and its control mice, Pbk$^{KI/KI}$ and Pbk$^{WT/WT}$ mice were isolated according to standard methods by collagenase digestion through the common bile duct (Dhawan *et al*, 2015). Islets were then separated from digested exocrine tissue by hand-picking following a Histopaque-1077 (Sigma-Aldrich) gradient. The isolated islets were allowed to recover in media (RPMI1640 with 10% FBS, 1% SP, 1% glutamine, and 10 mM glucose) for 2 h and were used for subsequent experiments.

Human islets were obtained from the pancreas of multiple human donors through Prodo Laboratories Inc (USA). Details for human donor islets used in this study are listed in Appendix Table S1. The human islets were cultured in a recovery culture media (Prodo Laboratories Inc, USA) with 5% human AB serum, 1% Glut supplement, and 1.2% Triple antibiotic supplement for 24 h to let them recover and adjust to the *in vitro* incubation environment. Informed consent was obtained from all human subjects and that the experiments conformed to the principles set out in the WMA Declaration of Helsinki and the Department of Health and Human Services Belmont Report.

### Development of gene knock down and overexpression cell lines

A lentiviral packing system including plasmids pMD2.G and psPAX2 (Sigma) was utilized to knock down gene expression via delivering shRNA into the target cells. All lentiviral shRNA plasmids were derived from a pLKO.1-puromycin backbone. shRNA plasmids of JunD, Pbk, and scramble shRNA were obtained from the University of Pennsylvania Perelman School of Medicine High-Throughput Screening Core. Specific shRNA sequences can be found in Appendix Table S3.

LentiCRISPRv2 was obtained as a gift from Dr. Anil Rustgi for use to knock out genes via a CRISPR technique. The lentiCRISPRv2 vector with *Men1* targeted sgRNA was sequenced to confirm that the correct insert was in place. Specific sgRNA sequences are in Appendix Table S3. To produce lentivirus, HEK293T cells were transfected with pMD2.G, psPAX2, and the sgRNA expression plasmid of interest using the calcium chloride precipitation method as previously reported (Katona *et al*, 2019).

A pMX-puro-Pbk construct, as previously published, and a mutant plasmids with K64K65 to AA based on pMX-puro-Pbk construct, were transfected into INS-1 cell line to induce ectopic expression of Pbk.

Cells were transduced by the virus of choice in the presence of 4 μg/ml polybrene (hexadimethrine bromide) for lentivirus infection. 24 h after completion of transduction, cells were then selected with puromycin for 72 h.

### Cell growth curve and MTS assay

The different cell lines were seeded at $1–2.5 \times 10^5$ cells per well in triplicate in 12-well plates and cells were counted at the indicated days.

As for the MTS assay, PIME cell lines were plated in a 96-well plate at a density of $1 \times 10^4$ cells/well. After overnight incubation, the cells were treated as described for the indicated time. The MTS [3-(4, 5-Dimethylthiazol-2-yl)-5-(3-carboxymethoxyphenyl)-2-(4-sulfophenyl)-2H-tetrazolium] assay kit (Promega) was utilized to

assess cell growth and was performed according to the manufacturer's instructions. Absorbance of each well was measured at 490 nm using an ELISA plate reader (SpectraMax M2, Molecular Devices, USA).

## Western blot analysis

Cultured cell lines and primary mouse or human islets were collected and lysed with RIPA buffer containing protease and phosphatase inhibitors. Protein concentrations were determined using a BCA assay kit (Thermo Scientific). Cell lysates were subjected to polyacrylamide gel electrophoresis on 4–12% Bis-Tris Plus Blot SDS–PAGE gels (GenScript), and protein was transferred to PVDF membranes (Life Technologies). Blocking was performed in TBST containing 5% non-fat dry milk based on the manufacturer's blocking instructions. The details of antibodies including first antibodies and second antibodies used for WB assay in this study are listed in Appendix Table S3. All primary antibodies for WB were used with same dilution of 1:1,000, except the Mouse monoclonal anti-beta actin, with 1:10,000 dilution. The corresponding second antibodies including goat Anti-Rabbit IgG (H + L)-HRP and goat Anti-Mouse IgG (H + L)-HRP were used with 1: 10,000 dilution. The proteins were visualized by detection with Amersham ECL Western blotting detection reagents (GE Healthcare). The information of all reagents and commercial kits are also listed in Appendix Table S3.

## Co-immunoprecipitation (co-IP) from HEK293T cells

HEK293T cells were transfected using calcium phosphate method. Two days after the transfection, the cells were lysed for 5 min at 4°C in 50 mM Tris–HCl, pH 7.4, 0.8% NP40, 10% glycerol, 150 mM NaCl, containing complete protease inhibitors (Roche), and PhosSTOP (Roche) and cleared by centrifugation. 20 µl of anti-Flag M2 Magnetic Beads (Sigma) equilibrated with Lysis buffer was then added, and the mixture was incubated for 60 min. The beads were washed five times with lysis buffer, and bound proteins were eluted in sodium dodecyl sulfate (SDS) sample buffer and separated by SDS-polyacrylamide gel electrophoresis for immunoblotting.

## Gene expression analysis by quantitative RT–PCR

RNA was extracted from cultured cells with TRIzol followed by subsequent isolation using an RNeasy Mini Kit (Qiagen). RNA (1.0 µg) was reverse transcribed into cDNA and real-time PCR (RT–PCR) was performed using a Quantitative SYBR-Green PCR Kit (Qiagen) and a 7500 Fast Real Time PCR System (Applied Biosystems). Sequences of primer sets used can be found in Appendix Table S3. Transcription levels were normalized to actin.

## Chromatin immune precipitation (ChIP) assay

ChIP assays were performed using a Quick CHIP kit (Novus Biologicals) according to the manufacturer's instructions. Approximately $5 \times 10^6$ cells were used for each immunoprecipitation. Briefly, cells were fixed with 1% formaldehyde and lysed in a ChIP lysis buffer with protease inhibitors, and cellular DNA was sheared with sonication. Then, the lysate was incubated with either control IgG or a specific primary antibody (4 µg) at 4°C overnight, and collected

with protein G agarose beads. The protein–DNA complexes were eluted from the beads and incubated at 65°C overnight to break the protein–DNA crosslinking. DNA was amplified by real-time PCR using primer pairs detecting indicated genes and SYBR-Green reaction mix (Qiagen). Specific primer sequences, reagents, and commercial kits can be found in Appendix Table S3. Each assay was implemented in triplicates with the mean normalized to input chromatin and reported as percent input.

## Luciferase report system and luciferase assay

HEK293T cells were transiently co-transfected with pcDNA3.1-JunD, pCDNA3.1-Menin, pCDNA3.1 vector, pRL-TK (internal control reporter vector), and either pGL3-based wild-type or mutant predicted JunD binding site-driven constructs. 36 h later, cells were lysed and luciferase activity was measured using the Dual-Glo Luciferase Assay System (Promega) following the manufacturer's instructions. The relative luciferase activity was determined by the ratio between firefly luminescence and renilla luminescence.

## Protein purification

To express the recombinant Erk2 protein, Erk2 expression plasmid was transformed into Rosetta(DE3)pLysS competent *E. coli* cells (Millipore Sigma), which were grown to an $OD_{600}$ of ~0.7 and induced with 0.5 mM of Isopropyl β-D-1-thiogalactopyranoside (IPTG) at 16°C for 16 h. Cells were harvested by centrifugation for 20 mins at 4500 rpm and lysed by sonication in a lysis buffer (pH 8.0) containing 25 mM Tris, 300 mM NaCl, 0.1 mg/ml phenylmethylsulfonyl fluoride (PMSF). After centrifugation, the supernatant was isolated and passed over Ni-resin (Thermo Scientific), which was subsequently washed with 10 column volumes of lysis buffer supplemented with 25 mM imidazole and 10 mM 2-mercaptoethanol (βME). Protein was eluted with buffer (pH 8.0) containing 25 mM Tris, 200 mM NaCl, 200 mM imidazole, 10 mM βME, and dialyzed into buffer (pH 7.0) containing 25 mM HEPES, 200 mM NaCl, and 10 mM 2-mercaptoethanol overnight. After dialysis, proteins were aliquoted, snap-frozen in liquid nitrogen, and stored at 80 °C for further use.

## *In vitro* Pbk kinase activity assay

The *in vitro* kinase assay was performed as previously reported (Xing *et al*, 2013). Briefly, 100 ng of purified recombinant His-Erk2 (purified using the method in Appendix) were incubated with 0.15 µg of recombinant active PBK or IP-ed Pbk from whole cell lysate in kinase buffer (35 mM Tris–HCl pH 7.5, 10 mM $MgCl_2$, 0.5 mM EGTA, 0.1 mM $CaCl_2$,) containing 100 µM ATP for 30 min at 30°C in a final volume of 40 µl. Reactions were terminated by addition of concentrated sample buffer and separated by SDS–PAGE followed by exposure to X-ray film.

## Blood glucose test

To measure blood glucose number, 1 mm of tail was clipped and a drop of blood was tested with a glucometer and disposable test strips (One touch, USA). Mice were considered hyperglycemic (diabetic) with a single blood glucose measurement of > 150 mg/dl for fasting blood glucose or two consecutive measurements > 250 mg/dl for random blood glucose.

## Intraperitoneal glucose tolerance test (IPGTT)

The intraperitoneal glucose tolerance test (IPGTT) was performed on different mice in this study to evaluate their glucose tolerance. The mice were fasted overnight (16 h) before the start of the assay. Baseline blood glucose was measured from tail vein blood using glucometer (time 0). Mice were then administrated with glucose (2.0 g dextrose/kg body weight) by IP injection and blood glucose was measured at 15, 30, 60, 90, and 120 min after injection from the tail vein.

## Food intake measurement

Body weight was measured (to the nearest 0.1 g) at the beginning and end of the 5-day test. The average of these measurements was used for all subsequent analyses. For each mouse, average daily food intake was calculated. Intakes adjusted for body weight were derived by dividing average intake by the average body weight. The test was repeated three times in first three weeks with MI administration on HFD-induced Diabetes mice, once for each week.

## Adiposity index calculation

Adiposity index was determined by the weight of epididymal divided by body weight × 100, and expressed as adiposity percentage.

## Flow cytometry to test cell cycles

Pbk ectopic expression PIME cells and control cells were seeded at a density of $4 \times 10^5$ cells in a 10 cm$^2$ plate and cultured for 24 h. Cells were then harvested by trypsinization and fixed using 70% ethanol and stored at $-20°$ C until further processing. For the flow cytometry assay, the fixed cells were centrifuged, rinsed in PBS, suspended in 500 μl PBS containing 50μg/ml propidium iodide (PI) and 100 μg/ml RNase A, and examined by flow cytometry (BD accuri C6 flow cytometer, BD Biosciences, USA). Data were analyzed using FlowJo software. All experiments were carried out three times. Each histogram was constructed from at least 10,000 cells.

## Insulin tolerance test (ITT)

After a 6-h fast, mice were given an IP injection of insulin (0.75 U/kg) delivered in 1 ml/kg saline. Blood glucose was measured at baseline (0), 15, 30, 45, and 60 min after injection with a hand-held glucometer and disposable test strips (One touch, USA).

## GSIS assays *ex vivo*

Insulin release function of the isolated mouse islets in response to glucose stimulation was tested by Islet Cell Biology Core at UPenn using an automated perfusion system. After two days of incubation in PRMI1640 media with 10% FBS, 1% SP, 1% glutamine, and 10 mM glucose at 37°C with 5% $CO_2$, islets were moved into the perfusion system and subjected to 0, low (3 mM), and high (16.7 mM) glucose concentration at 0, 30, and 50 min, respectively.

The system was switched back to 0 glucose at 70 min followed by KCL (30 mM) stimulation at 90 min for another 20 min. Insulin was harvested automatically and measured by Radioimmunoassay and Biomarkers Core at UPenn using the ELISA method.

## Immunofluorescence staining and imaging of pancreas sections

Pancreas samples were dissected, fixed in 4% paraformaldehyde in PBS at room temperature, and embedded in paraffin. Paraffin embedded sections of pancreas samples were then deparaffinized and antigen retrieval was performed in Tris-EDTA buffer (pH 9.0) using a high presser oven for 2 h. The sections were blocked for 1 h at room temperature using a blocking buffer (5% goat serum, 0.05% Tween-20 in PBS) followed by incubation with primary antibodies overnight at 4°C. All primary antibodies for IF including anti-mouse Insulin, Pbk, Ki67, and BrdU antibodies were used with same dilution of 1:250. After washing three times with PBS, sections were then treated with corresponding secondary antibodies (1:200 dilution for all second antibodies used for IF) and DAPI (Roche) for 1 h at room temperature. The details of the antibodies used in this study can be found in Appendix Table S3. The sections were visualized using a Nikon Eclipse E800 fluorescent microscope with a CCD digital camera.

To avoid bias during imaging and selecting sections, the samples were blinded by randomly numbering samples to avoid group related information, prior to moving to next step of analysis including staining the sections and collecting the image. The quantification analysis of the positive staining images was carried out using ImageJ software.

## Beta cell mass calculation

First, the percentage of beta cell area within the pancreas from six different sections of each mouse pancreas was measured. Beta cell area was determined by insulin staining area observed using a microscope (DMI6000B, Leica) with whole section scanning function. The percentage of beta cell area over whole pancreas section was calculated using ImageJ software. The six different sections from each pancreas sample were sequentially cut with each section separated by at least 250 μm. Then, the beta cell mass was calculated by multiplying the average percentage of beta cell area by the pancreas weight of corresponding animals.

## Proliferation index analysis

BrdU$^+$ or Ki67$^+$ β cells were assessed by immunofluorescence microscopy. Insulin$^+$ cells showing nuclear DAPI staining were considered as β cells. Insulin$^+$ cells showing nuclear co-localized staining for DAPI$^+$ and BrdU$^+$ (or Ki67$^+$) were counted as proliferating β cells. The proliferation index for each islet was determined with the score from the number of double-positive cells (Ins$^+$/BrdU$^+$ or Ins$^+$/Ki67$^+$) over the number of whole insulin$^+$ cells with DAPI staining.

## RNA-seq data analysis

RNA-seq reads (PRJEB30761) were downloaded from European Nucleotide Archive database (www.ebi.ac.uk/ena/). FastQC was used to assess sequencing reads qualities and then Trimmomatic v.0.39 was used to trim adapters and low-quality ends until all reads scored

**The paper explained**

**Problem**

An insufficient number of functional pancreatic beta cells is a central feature of both type 1 and advanced type 2 diabetes. As such, there has been considerable interest in exploring strategies for inducing beta cell regeneration as a novel mechanism for diabetes treatment. Pancreatic beta cells undergo compensatory proliferation in the early phase of obesity or high fat diet (HFD)-induced type 2 diabetes (T2D), but is underexplored. Understanding the underlying mechanism of compensatory beta cell proliferation as well as the regulation of this proliferation is crucial and may lead to improved treatments for diabetes.

**Results**

Pbk was identified as a key protein for mediating HFD-induced compensatory beta cell proliferation using a Pbk kinase dysfunctional mouse model. Mechanistically, the transcription factor JunD recruits menin and HDAC3 complex to the Pbk promoter to reduce histone H3 acetylation, leading to epigenetic repression of Pbk expression. Pharmacologically blocking the menin–JunD interaction menin inhibitors (MIs) increased compensatory beta cell proliferation, resulting in both improved hyperglycemia and glucose tolerance in HFD-induced diabetic mice, demonstrating the key impact of MIs on influencing expression of Pbk and beta cell proliferation in mouse models.

**Impacts**

The current studies illustrate the novel role of the menin/JunD/Pbk axis as a key regulator of compensatory beta cell proliferation, unraveling a novel approach to enhancing beta cell proliferation by regulating the menin pathway through administering MIs. These novel findings highlight the potential to further develop MIs to increase Pbk expression and beta cell regeneration, which is a novel mechanism of action for developing future islet function-enhancing drugs for improving treatment of diabetes.

above 20 at each position (Bolger *et al*, 2014). RNA-seq reads were aligned to human GRCH37 genome assembly (hg19) using RSEM (Li & Dewey, 2011) with default parameters. Only unique mapped reads were considered for further analysis. Normalized expression value, fragments per kilobase of exon per million reads mapped (FPKM), was calculated for each gene using StringTie (Pertea *et al*, 2015). The gene was further considered expressed if its expression value is greater than 1 in at least one subject. For differential expression analysis, read counts were measured within Ensembl genes (GRCH37) using featureCounts, and then, edgeR (Robinson *et al*, 2010) was used with adjusted $P$ value $\leq 0.05$ and fold change $> 2$ from control LFD against hyperglycemic HFD samples.

**Statistical analysis**

For all experiments, the number of biological replicates (n), measure of central tendency (e.g., average), error bars, and statistical analysis has been explained in the Figure legends. All statistically significant comparisons are indicated in the Figures and corresponding legends. Data are shown as mean $\pm$ SEM. GraphPad Prism software and R 3.5.0 (http://www.R-project.org/) were used for statistical analysis. Student's *t*-test and ANOVA were used to determine the significance of the results. $*P < 0.05$, $**P < 0.01$, $***P < 0.001$, exact P value could be found in the Figure legends and the source data.

## Data availability

This study includes no data deposited in external repositories.

**Expanded View** for this article is available online.

## Acknowledgements

We thank the processing of the samples for histological studies from the Molecular Pathology & Imaging Core at University of Pennsylvania, the isolation of mouse islets at Pancreatic Islet Cell Biology Core at University of Pennsylvania, and the generation of gene mutant KI mice at CRISPR/Cas9 Mouse Targeting Core and Transgenic and Chimeric Mouse Facility at University of Pennsylvania. This work was supported by Harrington Discovery Institute Innovator Scholar Award, and a Sanofi Innovation Award (iAward).

## Author contributions

Study conception and design and the analysis and interpretation of the data: XH and JM; Manuscript writing: JM and XH; Manuscript revision: All other authors; *In vitro* and *in vivo* experiments: JM; *In vitro* kinase assay: BX; Some Western blots and some animal studies: YC; Cell cycle studies: XH; Tissue section preparation and part of histology studies: KEB; Bioinformatics analysis and plotting figures: CT; Purification of Erk protein for *in vitro* kinase study: SD; Animal experiments and tissue harvesting: CA, TH, SL, GX, and YW; Reviewing data and providing advice: ZF and BK; Suggestion for animal number used in experiments and data statistical analysis: YR and HL; Supervision of primary islet-related experiments: MY and AN. Final approval of the paper: All authors.

## Conflict of interest

The authors declare that they have no conflict of interest.

## For more information

https://www.ebi.ac.uk

https://www.proteinatlas.org

https://www.med.upenn.edu/apps/faculty/index.php/g275/p10254 (The authors' website).

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
