## [Review Process File · EMBO Molecular Medicine]

Menin-regulated Pbk controls High fat diet-induced compensatory beta cell proliferation

Jian Ma, Bowen Xing, Yan Cao, Xin He, Kate Bennett, Chao Tong, Chiying An, Taylor Hojnacki, Zijie Feng, Sunbin Deng, Sunbin Ling, Gengchen Xie, Yuan Wu, Yue Ren, Ming Yu, Bryson Katona, Hongzhe Li, Ali Naji, and Xianxin Hua

DOI: [10.15252/emmm.202013524](https://doi.org/10.15252/emmm.202013524)

Corresponding author: Xianxin Hua (huax@pennmedicine.upenn.edu)

Review Timeline:

Submission Date:	28th Sep 20
Editorial Decision:	14th Oct 20
Revision Received:	27th Jan 21
Editorial Decision:	3rd Feb 21
Revision Received:	7th Feb 21
Accepted:	12th Feb 21

Editor: Zeljko Durdevic

Transaction Report:

14th Oct 2020

Dear Dr. Hua,

Thank you for the submission of your manuscript to EMBO Molecular Medicine. We have now received feedback from the three reviewers who agreed to evaluate your manuscript. As you will see from the reports below, the referees are overall supportive of the study but also raise some concerns that should be addressed in a major revision. Particular attention should be given to measuring the Pbk activity using kinase assay, performing statistical analyses using analysis of variance (ANOVA) test, determining statistical power of the sample size and improving data presentation as suggested by the referee #1. Please revise the title of the manuscript to better highlight the main conclusion of the study. Additional experiments towards elucidating the mechanism of Pbk-mediated control of beta cell proliferation are welcome, however not required for further consideration of your manuscript in our journal.

Addressing the reviewers' concerns in full, experimentally or in writing, will be necessary for further considering the manuscript in our journal, and acceptance of the manuscript will entail a second round of review. EMBO Molecular Medicine encourages a single round of revision only and therefore, acceptance or rejection of the manuscript will depend on the completeness of your responses included in the next, final version of the manuscript. For this reason, and to save you from any frustrations in the end, I would strongly advise against returning an incomplete revision.

We would welcome the submission of a revised version within three months for further consideration. However, we realize that the current situation is exceptional on the account of the COVID-19/SARS-CoV-2 pandemic. Please let us know if you require longer to complete the revision.

I look forward to receiving your revised manuscript.

Yours sincerely,

Zeljko Durdevic

***** Reviewer's comments *****

Referee #1 (Remarks for Author):

Ma and colleagues present data suggesting that the PDZ-binding kinase (Pbk) regulates increased pancreatic beta cell proliferation via control of the menin/JnD transcriptional complex. These data

are interesting and the experiments are well presented. I have the following comments:

1. Throughout the paper, there is no mention of what the substrates for Pbk are-this lack of mechanistic insight undermines the impact of the paper. For example, could menin or JunD be substrates for Pbk as part of a negative feedback loop?
2. A major concern is the incorrect statistical analysis applied to most of the data-comparison of multiple groups requires ANOVA with post hoc testing. Instead, the authors have used unpaired t-test throughout. This is inappropriate for any experiments with more than 2 groups.
3. The new gold standard in data presentation is to show the mean values as well as the replicates in the graphs, rather than a solid bar with errors
(<https://rupress.org/jcb/article/219/6/e202001064/151717/SuperPlots-Communicating-reproducibility-and>)
The authors do not present their data correctly in many cases, for example Fig. 1E, this is n=4 mice per group with 10 islet images per mouse. The authors have plotted each islet as if it were one animal, and this is inappropriate and needs to be corrected.
4. There are no power calculations to indicate how the authors arrived at n=4 mice per group as being appropriate. This seems very low to me, and barely above the minimum number of animals needed for an in vivo experiment. The authors need to justify such a low n number in their experiments.
5. Rather than RNA levels or protein expression, I would like to see Pbk kinase assays to support many of the conclusions in the paper-why were these not carried out?
6. Fig. 2J-this GTT looks strange-why does the difference seem to disappear at 12 weeks?
7. Fig. 3C versus 3G-why is there a difference in the basal expression level of Pbk in the PIME cells? Fig. 3C-undetectable, Fig. 3G, detectable?
8. Fig. 5 E, F-the use of % Input is vague and uninformative here-please clarify for the reader what is being plotted. Fig. 5F is only n=2-any reason why this was not completed n=3?
9. Fig. 6-it would have increased the impact of the paper if the authors had demonstrated that the kinase activity of Pbk was increased upon treatment with the Menin inhibitor.
10. Given how few cells stain positive for Pbk in the islets (Fig. 1, 2, 7 8), this would suggest that Pbk is not a major driver of beta cell proliferation in the various HFD or diabetic states-the authors should comment on this.
11. Fig.1-single cell sequencing rather than transcriptomic analysis would have been more informative here.
12. I could not see any description of blinding of sections prior to quantitation, an essential step to avoid any degree of bias-please clarify.

Referee #2 (Remarks for Author):

In this elegant and thorough manuscript, Ma and colleagues explore the Menin/JunD/Pbk axis in beta cell proliferation. They comprehensively show that Pbk is induced in islets by high fat diet (HFD) conditions in mice and human islets, and that Pbk is normally repressed in beta cells by the recruitment of a Menin-HDAC3 complex by JunD to the Pbk promoter. Using an extensive array of methods and models, including several cell lines and whole animal models, the authors convincingly demonstrate that disruption of this recruitment event leads to increase in Pbk transcription and improved beta cell proliferation in a Pbk kinase-activity-dependent manner. Finally, the authors show that application of small molecule inhibitors design to block Menin binding to JunD improves beta cell mass and glucose tolerance in a HFD-induced murine model of type 2 diabetes.

While the effects of most experiments are mostly moderate and only work in males, the results of this extensive and multi-method approach, taken together, make the case of the Menin/JunD/Pbx axis very convincing. The manuscript and the data presented tell a complete and convincing story, and besides some random typos ("custered" instead of "clustered" in line 119, "fos" instead of "for" in line 188, "coin" instead on "corn" in line 885), this reviewer does not see a need for any further revisions.

Referee #3 (Comments on Novelty/Model System for Author):

This study has used comprehensive and complementary approaches including Pbk kinase dead knockin mice, Menin conditional knockout cell lines, JunD shRNA knockdown, inhibitors of Pbk, AP1, Menin, and HDAC, to demonstrate their points.

Referee #3 (Remarks for Author):

This manuscript addressed the role of Pbk, a serine/threonine protein kinase, in compensatory beta cell proliferation. Using many complementary approaches including Pbk kinase dead knockin mice, Menin conditional knockout cell lines, JunD shRNA knockdown, inhibitors of Pbk, AP1, Menin, and HDAC, this study gracefully demonstrated that (1) JunD first bound to the promoter of Pbk and recruited menin and HDAC3; (2) the JunD/menin/HDAC3 complex then reduced transcription-activating acetylated histone3 level at the Pbk promoter and the expression of Pbk proteins; and (3) high-fat diet increased Pdk+/insulin+ cells from nearly 0% to ~5% in the mouse islets. The manuscript was well-written with strong scientific rigor and high impact.

Minor concerns:

1. The title needs to be revised because Pdk did not control menin/JunD/Pbk axis.
2. The data in Figure S1A and B are from public datasets, and need references.
3. It is important to document that the proliferating cells in the islet were Pbk+ in Figure 1D, 1G, and 8E. It helps define whether the compensatory beta cell proliferation effect of Pbk is cell-autonomous or not.
4. Why did ectopic expression of Pbk increase the sensitivity of cells to the Pbk inhibitor?
5. Why did PBK kinase dead knockin only impair glucose tolerance in male mice, but not in female mice? Can the authors provide any thoughts in discussion? Any correlation in human male vs female patients?
6. In the result section regarding Figure 2F-G, please specify that these were male mice.
7. Unlike Fig. 4L, Fig. S5C did not show AP1 inhibitor induced Pbk expression in a dose-dependent manner (no induction at 0.1 and 1 uM concentration). Please revise the result.

8. Can the authors provide any thoughts on how high fat diet affected the recruitment of JunD/menin/HDAC3 on the Pbk promoter?
9. Can the authors provide explanation why there was no Pbk in lane 1 of Fig. 3C, but Pbk can be detected in lane 1 of Fig. 4B?
10. In line 204, it should be Fig. 4D-F, not Fig. 1D-F.
11. In line 314, it should be oncogenic, not ocogenic.
12. Please verify the y-axis unit of the ChIP figures.

Thank you for the opportunity to revise and improve our manuscript. We appreciate the insightful comments and suggestions from the reviewer to improve our manuscript. We have performed multiple new experiments to address the reviewers' comments and concerns, besides revising the other relevant points. All changes made in the manuscript are highlighted. We hope that you and the reviewers would agree that we have addressed their comments and concerns satisfactorily in the revised manuscript. Below are point-by-point responses to the comments and suggestions:

Referee #1 (Remarks for Author):

Ma and colleagues present data suggesting that the PDZ-binding kinase (Pbk) regulates increased pancreatic beta cell proliferation via control of the menin/JunD transcriptional complex. These data are interesting, and the experiments are well presented. I have the following comments:

1. Throughout the paper, there is no mention of what the substrates for Pbk are-this lack of mechanistic insight undermines the impact of the paper. For example, could menin or JunD be substrates for Pbk as part of a negative feedback loop?

Response: We appreciate these insightful and constructive questions from the reviewer. Pbk/TOPK was shown to phosphorylate several protein substrates including p53 related protein kinase (PRPK) in colorectal cancer cells (Zykova *et al*, 2017) and Cdk1/cyclin B1/PRC1 kinase substrate complex (Abe *et al*, 2007), correlating with its role in promoting cell cycle progression. However, thus far, there have been no definite published data showing the key substrates that mediate Pbk's function in promoting cell cycle progression. The reviewer raised the intriguing question regarding the potential feedback effect of Pbk on phosphorylating menin and/or JunD, and if Pbk indeed phosphorylates one or both proteins, the results will provide further insights into feedback loop among the three proteins.

As previous studies report that Pbk is a member of the MAPKK family (Abe *et al*, 2000; Gaudet *et al*, 2000; Matsumoto *et al*, 2004) and our data also show upregulation of phosphorylated Erk1/2 in Pbk ectopically expressed PIME or INS-1 cells (Fig.1A here in this letter, and this was adapted from Fig EV1C), **firstly**, we performed *in vitro* kinase activity assay to determine whether Pbk can phosphorylate a substrate *in vitro*. To this end, we expressed Erk2 in *E coli* in pMCSG7 vector, and purified Erk2 protein using the Ni-NTA column. The purified protein was in the expected size on Coomassie blue-stained gel (data not shown).

We obtained commercially available and purified Pbk protein (Cat No: MBS145337, MyBioSource, San Diego, CA, USA) and performed the *in vitro* kinases assay with Pbk and Erk2, in the buffer with or without ATP. The results show that incubation of Pbk with Erk2 in the presence of ATP indeed induced Erk2 phosphorylation as shown in Western blot with the

phosphorylation-specific anti-pERK1/2 antibody (Cat No: #9101, Cell Signaling Technology, USA) (Fig. 1B, here in this letter, lane 3). In contrast and as expected, removal of either Pbk or ATP from the reaction led to abolishing of Erk2 phosphorylation (Fig. 1B, lane 1 and 2, respectively). On the other hand, addition of various concentrations of menin inhibitor (MI) did not affect the PBK-mediated phosphorylation of Erk2 *in vitro* (Fig.1B, lanes 4-5). As a control, the amount of purified Erk2 was comparable among each of the reactions (Fig. 1B, lanes 1-5, lower panel).

In addition, to confirm the kinase activity of endogenously expressed Pbk on Erk2, we immunoprecipitated (IP-ed) Pbk from WT PIME cells or menin knock-out (KO) PIME cells, which showed higher Pbk expression as compared to the control WT cells in Western blot (Fig.1C, top panes Input-Pbk. We next used the IP-ed Pbk (IP: Pbk) from the same amount of cell lysates (PIME cell lysates protein concentration were measured by BCA) from either WT or KO PIME cells to perform *in vitro* kinase assay with the purified Erk2 as substrate, followed by Western blot to detect the phosphorylated Erk2. The results showed that a larger amount of Pbk is associated with the higher level of the phosphorylation of the substrate, Erk2 (Fig.1C). Together with the results from Figure 1A-B, these findings suggest that menin deletion-induced expression of Pbk results in upregulation of Erk2 in the cells. It is known that Erk2 contributes to enhancing cell proliferation. Collectively, these results suggest that Erk2 is one of substrates of Pbk kinase. We have added these new data (Fig 1B and C) into the updated Fig.1 in the revised manuscript as Fig.1P and 1Q, and incorporated the related description in the revised manuscript in line 126-141.

Fig.1. Pbk phosphorylates Erk2. **(A)** The effect of Pbk overexpression on phosphorylation levels of Erk1/2. WT PIME cells transduced with vector or Pbk-expressing lentivirus were processed for Western blot analysis with the indicated antibodies. **(B)** Purified recombinant His-Erk2 protein (100 ng) incubated with the purified PBK (0.15 μ g) in the presence or absence of ATP (100 μ M) or menin inhibitor (1 μ M or 0.1 μ M), as indicated. The final volume is 40 μ l. After SDS-PAGE, the phosphorylation of Erk2 was detected with the pERK1/2 antibody or separately with the anti-Erk antibody. Anti-ERK1/2 antibody showed that equal amount of Erk2 was used for the kinase assay. **(C)** Purified recombinant His-Erk2 proteins were incubated with IP-ed Pbk from PIME cells (WT or menin^{-/-}) for the kinase activity assay. The phosphorylation of Erk2 was detected with the anti- pERK1/2 antibody. The Western blot incubated with the anti-ERK1/2 antibody showed that an equal amount of ERK2 was loaded for the kinase assay.

Secondly, as we showed in the previously submitted supplemental figure, and also illustrated in Figure 1 to show the impact of ectopic Pbk expression in PIME cells on Erk phosphorylation, we also observed the enhanced JunD phosphorylation in the cells in the same experiment, as

illustrated here in Figure 2 (Fig. 2A, which was adapted from Fig EV1C) (at the 100th serine, anti-pJunD, CST, USA). To further determine whether JunD could be a substrate of Pbk, we also generated expression constructs to express JunD in *E. coli* and expressed and purified the JunD protein. We co-incubated Pbk with the purified JunD, as we performed Pbk/Erk2 *in vitro* kinase assay, and we confirmed that the anti-pJunD antibody worked as expected (Fig. 2A). Unfortunately, two independent *in vitro* kinase assays with JunD as the substrate failed to detect any JunD phosphorylation *in vitro* (data not shown), even we were able to detect robust Erk2 in the same experimental conditions, as shown in Figure 1A, as well as JunD phosphorylation as shown in Figure 2A.

Then, we sought to determine whether co-express Pbk and JunD in HEK293 cells we co-transfected Pbk cDNA-expressing plasmid with JunD cDNA-expressing plasmid into 293 cells, followed by Western blot to detect the potential JunD phosphorylation. The results indicate that the transfected JunD had substantial background phosphorylation (Fig 2B, lane 2). However, co-transfection with Pbk-expressing plasmid did not affect levels of the JunD phosphorylation (Fig. 2, lane 3). Moreover, we used IP-ed Pbk (IP: Pbk) from the same amount PIME cell lysates (with or without menin) and purified JunD protein from *E. coli* to perform the *in vitro* kinase assay. From the WB results (Fig 2C), IP-ed Pbk from the cells did not influence phosphorylation of JunD *in vitro*. Collectively, these data indicate that JunD may not be a direct substrate of Pbk kinase.

Fig. 2. JunD may not be a direct substrate of Pbk. **(A)** The effect of Pbk overexpression on phosphorylation levels of JunD. **(B)** HEK293 cells cotransfected with HA-JunD and V5-PBK (WT or mutant) were lysed and subjected to Western blot using the indicated antibodies. **(C)** Purified recombinant His-JunD protein was incubated with IP-ed Pbk from PIME cells (WT or menin^{-/-}) for the kinase activity assay. Through running SDS-PAGE, the phosphorylation of JunD was detected with the pJunD antibody. Blot incubated with the anti-JunD antibody showed that the equal amount of JunD was used for the kinase assay.

Thirdly, according to the reviewer’s suggestion, we also performed *in vitro* phosphorylation of purified menin with purchased Pbk (MyBioSource, San Diego, CA, USA), in conditions as described in Figure 1 for Erk2 phosphorylation. However, the results did not detect any obvious phosphorylation, using the anti-phosphorylated menin antibody (Xing *et al*, 2019) (Fig. 3). As a positive control, we showed that our antibody specifically recognized the phosphorylated menin that was induced by Forskolin treatment (Fig. 3, lane 7 vs 6). Of course, we are fully aware that these negative results do not rule out the possibility that inside cells Pbk may be still able to directly or indirectly phosphorylate menin and/or JunD or phosphorylate these proteins in other sites that could not be detected by our antibodies, to modulate their function, or we might not

have gotten the best assay conditions to demonstrate the phosphorylation. On the other hand, we do not have the definitive evidence for now to show that Pbk can directly phosphorylate Menin and JunD.

Fig.3. Menin may not be a direct substrate of Pbk. Purified recombinant menin proteins were incubated with active Pbk in the presence or absence of menin inhibitor (1 μ M or 0.1 μ M). After SDS-PAGE, the phosphorylation was detected with the anti- pSer487 menin antibody. Anti-menin antibody showed that equal amount of menin was used for the kinase assay. For lanes 6-7, whole cell lysate from PIME (treated with or without 10 μ M Forskolin for 30 min) cells was also blotted with the same pSer487 menin antibody, or the anti-total menin antibody as indicates.

To address the reviewer’s concern, we have added the following sentences in Discussion section from line 351 to 358: “While Pbk/TOPK was reported to phosphorylate several protein substrates including p53 related protein kinase (PRPK) in colorectal cancer cells (Zykova *et al.*, 2017) and Cdk1/cyclin B1/PRC1 kinase substrate complex (Abe *et al.*, 2007), our preliminary *in vitro* studies did not detect obvious phosphorylation of menin or JunD by Pbk using the anti-menin S487 phosphorylation antibody or anti-JunD S100 antibody, respectively. Rather, we found the purified Pbk as well as the *Men1* deletion-induced Pbk from cell lysates phosphorylated Erk2 in the *in vitro* kinase assay. However, the correlation between Pbk-enhanced Erk1/2 phosphorylation and beta cell proliferation needs further investigation.”

2. A major concern is the incorrect statistical analysis applied to most of the data-comparison of multiple groups requires ANOVA with post hoc testing. Instead, the authors have used unpaired t-test throughout. This is inappropriate for any experiments with more than 2 groups.

Response: We appreciate the suggestion from the reviewer and have performed the statistical analysis for the experiments with more than two groups with the ANOVA method in the revised manuscript, as shown in Fig. 1L, 1M, Fig. 2 E, 2J, Fig.3O, Fig. 6E, 6F, 6G, Fig.7B, Fig. 8B, 8C, 8M, 8O, Fig. EV1A, Fig. EV2A-H, Fig.EV5C-I, 5L-M, and Appendix Fig.S3A-B.

3. The new gold standard in data presentation is to show the mean values as well as the replicates in the graphs, rather than a solid bar with errors (<https://rupress.org/jcb/article/219/6/e202001064/151717/SuperPlots-Communicating-reproducibility-and>). The authors do not present their data correctly in many cases, for example

Fig. 1E, this is n=4 mice per group with 10 islet images per mouse. The authors have plotted each islet as if it were one animal, and this is inappropriate and needs to be corrected.

Response: We reanalyzed the data, and presented the islet data from each mouse with a mean values, in distinct color, from four mice in updated Figure 1E, as well as in updated Figure 1I, Figure 2M, 2O, Figure 7M and N, Figure 8F, 8J, for the data from animal studies using the mean values as well as the replicates as reviewer suggested. We also revised the other relevant figures, including-Fig 1B-C,G, K, Fig 2F,G, N, Fig 3B,D, F, H, J, L-M, Fig 4D-K, Fig 5C-F, Fig 6B,D,I,L-M, Fig 7C-D, O, Fig 8G, K, N, P, Fig EV3B, Fig EV4B,D, and Fig EV5J-K, accordingly.

4. There are no power calculations to indicate how the authors arrived at n=4 mice per group as being appropriate. This seems very low to me, and barely above the minimum number of animals needed for an in vivo experiment. The authors need to justify such a low n number in their experiments.

Response: As the mice we used in each group were independent, we did a power calculation based on 4 mice and expected variations in each group when we designed the experiments. With n=4 in each group, we expected to detect an effect size of 2.39 (measured by mean difference/sd) with a power of .80 at the alpha level of 0.05. Based on this and also because we expected a large effect size, we designed our study only to detect such a large effect size. Moreover, in the JCB paper recommended by the reviewer, the authors also cited acceptable examples with n=3 in which multiple cells were measured in each sample (Figure 1 in the cited paper). To further strengthen our biostatistics analysis, we have also consulted and collaborated with Dr. Hongzhe Li, Professor of Biostatistics and Statistics, Director of Center for Statistics in Big Data, University of Pennsylvania, for further statistical analysis and presentation of the relevant data. He and a postdoctoral fellow in his group have now been added as new co-authors for the revised manuscript. Based on our experimental results, we did observe that tests on some of the parameters were significant based on the data we collected. So, we think that n=4 we used for our animal experiments in the study is acceptable for our statistical analysis.

5. Rather than RNA levels or protein expression, I would like to see Pbk kinase assays to support many of the conclusions in the paper-why were these not carried out?

Response: As detailed in response to #1 question, we have performed such studies and found that Erk2 can be phosphorylated *in vitro* by purified Pbk protein (Fig. 1), and also the menin deletion-induced Pbk IP-ed from cell lysates was correlated with increased Erk2 phosphorylation in the kinase assay (Fig. 1C). So, we believed that upregulated Pbk correlated with increased Pbk kinase activity.

In addition, we also tested whether MI-induced Pbk expression correlated with the enhanced Erk2 phosphorylation in cells. We stripped the blots from the Figure 6A, C, H, and K in manuscript, and used the recovered blots to test protein level of total Erk1/2 and phosphorylated Erk1/2. We incorporated the new obtained Erk1/2 and Phosphorylated Erk1/2 WB images into the original Figure 6 A, C, H, and K, and generated the updated Figure 4A-D here. We found that MI treatment increased Pbk expression, as well as Erk1/2 phosphorylation in a dose-dependent manner (Figure 4A here). Notably, the MI treatment failed to further increase Erk1/2 phosphorylation in the menin deleted cells (Fig. 4B here), highlighting the menin-dependence of the MI treatment-induced Pbk expression and increase of Erk1/2 phosphorylation in the cells. Importantly, the treatment of MI on primary mouse and human islets also led to the increasing of phosphorylated ERK level that correlates with the upregulated Pbk protein and mRNA level (Fig. 4C-D here). Together, these results, coupled with the *in vitro* Pbk/Erk2 kinase assay results

shown in Figure 1 (Updated Figure 1O and P in the revised manuscript) here, indicate that the induced Pbk expression level correlates with enhanced Pbk kinase activity inside cells.

Fig. 4. MI treatment increased phosphorylated ERK1/2. A, B. Menin WT PIME cells (A) and menin KO PIME cells (B) were treated with various doses of MI-463 for 48 hours, followed by detecting Pbk expression and Erk1/2 phosphorylation using WB. C, D. Primary mouse islets (C) and human islets (D) were treated with various doses of MI-463 for 5 days, followed by detecting PBK expression and ERK1/2 phosphorylation using WB.

6. Fig. 2J-this GTT looks strange-why does the difference seem to disappear at 12 weeks?

Response: During the 3-8 weeks, Pbk WT mice showed improved GT, likely via compensation from beta cell proliferation. However, the compensation effect could be reduced or even vanished if the mice were continuously under HFD stress prolonging to 12 weeks. We think that the difference disappeared in GTT, at least partly, resulted from failure of the compensation in the Pbk WT mice with a long time HFD stress.

7. Fig. 3C versus 3G-why is there a difference in the basal expression level of Pbk in the PIME cells? Fig. 3C-undetectable, Fig. 3G, detectable?

Response: Different exposure time for films in the Western blots can affect the signal levels in different films, making it difficult to compare the levels in Pbk basal expression between Fig.3C and Fig.3G. We have updated the Figure 3C with a long time-exposed image to avoid the confusion.

Fig. 4 (updated Figure 3C in revised manuscript) *Men1* knock out (KO) in PIME cells through Cre-mediated gene editing upregulated Pbk protein level.

8. Fig. 5 E, F-the use of % Input is vague and uninformative here-please clarify for the reader what is being plotted. Fig. 5F is only n=2-any reason why this was not completed n=3?

Response: We added the definition of “1% input” to Y axis label with in the first figure for chromatin immunoprecipitation (ChIP) assay (Figure 4D). For ChIP assay, the relative amount of the DNA bound by menin or other factors is expressed as the percentage of the amount of input DNA. As it is very sensitive and also only a very small percentage of the genomic DNA bound by transcriptional factors/chromatin-associated proteins is recovered during the ChIP, we took 1% of sonicated genomic DNA as the input DNA for all CHIP assays in this study to enlarge the relative ratio number. So, in the revised manuscript, in legend of the Figure, we added “1% input denotes the signal level normalized by the equivalent total input DNA (i.e. 1% of sonicated genomic DNA) for ChIP assay.” from line 995-997. And we also used “1% input” to replace “% input” in updated Figure 5E and 5F. In addition, to address the reviewer’s concern on N number in Figure 5F, we performed one more round independent CHIP assay to increase n number to 3 and updated Figure 5F with the new obtained data. As there are triplicate readings for each data point in each independent experiment.

9. Fig. 6-it would have increased the impact of the paper if the authors had demonstrated that the kinase activity of Pbk was increased upon treatment with the Menin inhibitor.

Response: We appreciate the reviewer’s comments and have performed the *in vitro* kinase activity assay with or without adding menin inhibitor to test whether MI owns ability to influence kinase activity of Pbk or not. The results as shown in Figure 1, as addressed at #1 extensively, indicate that MI treatment did not influence the kinase activity of Pbk on the substrate Erk2 *in vitro*, but increased Erk1/2 phosphorylation inside cells (Figure 4A here), and in a menin-dependent manner (Figure 4B here).

10. Given how few cells stain positive for Pbk in the islets (Fig. 1, 2, 7 8), this would suggest that Pbk is not a major driver of beta cell proliferation in the various HFD or diabetic states-the authors should comment on this.

Response: Pbk expression was upregulated in the islets of HFD or MI treated mice, albeit in a limited number of beta cells. Of note, Pbk is crucial for HFD or MI treatment-induced beta cell

proliferation because Pbk-KI abolished the MI treatment-induced beta cell proliferation (Figure 8D-F in manuscript). Although we could not rule out that other players are also important for HFD or MI-induced beta cell proliferation, our data showed that Pbk expression in a limited number of beta cells is crucial for HFD or MI treatment-induced beta cell proliferation, because Pbk-KI blocked HFD or MI treatment-induced beta cell proliferation. Moreover, upregulation of Pbk only in a limited number of beta cells could be a safe way to drive modest cell proliferation, yet not increase the high risk for beta cell tumorigenesis. To address the reviewer's comments, we added in Discussion section the following sentences in line 408 to 411: "Our data showed that Pbk upregulation in a limited number of beta cells is crucial for HFD or MI treatment-induced beta cell proliferation, as Pbk-KI blocked HFD or MI treatment-induced beta cell proliferations. Moreover, upregulation of Pbk only in a limited number of beta cells could be a safe way to drive modest cell proliferation, while not increasing beta cell tumorigenesis."

11. *Fig.1-single cell sequencing rather than transcriptomic analysis would have been more informative here.*

Response: We certainly agree with the reviewer that single cell sequencing rather than transcriptomic analysis would have been more informative, our group has not yet equipped with the capacity to perform such analysis. Nevertheless, our studies have already provided extensive data, ranging from cell biology studies, to pharmacological studies, and generating novel CRISPR-mediated Pbk KI and following phenotype analysis. These studies support our conclusion that Pbk is essential for HFD or MI-increased beta cell proliferation.

12. *I could not see any description of blinding of sections prior to quantitation, an essential step to avoid any degree of bias-please clarify.*

Response: We appreciate the reviewer's insightful comments and suggestion. In current study, most possible bias may happen when capturing and choosing fluorescence images to do quantification analysis to generate data in histology experiments. We took necessary measures to minimize bias. In the revised manuscript, we added the following section in the material and method section: " To avoid bias during imaging and selecting sections, the samples were blinded by randomly numbering samples to avoid group related information, prior to moving to next step of analysis including staining the sections and collecting the image." This description is now included in line 606 - 609 in revised manuscript.

Referee #2 (Remarks for Author):

In this elegant and thorough manuscript, Ma and colleagues explore the Menin/JunD/Pbk axis in beta cell proliferation. They comprehensively show that Pbk is induced in islets by high fat diet (HFD) conditions in mice and human islets, and that Pbk is normally repressed in beta cells by the recruitment of a Menin-HDAC3 complex by JunD to the Pbk promoter. Using an extensive array of methods and models, including several cell lines and whole animal models, the authors convincingly demonstrate that disruption of this recruitment event leads to increase in Pbk transcription and improved beta cell proliferation in a Pbk kinase-activity-dependent manner. Finally, the authors show that application of small molecule inhibitors design to block Menin binding to JunD improves beta cell mass and glucose tolerance in a HFD-induced murine model of type 2 diabetes.

While the effects of most experiments are mostly moderate and only work in males, the results of this extensive and multi-method approach, taken together, make the case of the Menin/JunD/Pbx axis very convincing. The manuscript and the data presented tell a complete and convincing story, and besides some random typos ("custered" instead of "clustered" in line 119, "fos" instead of "for" in line 188, "coin" instead on "corn" in line 885), this reviewer does not see a need for any further revisions.

Response: We appreciate the positive comments on our manuscript from the reviewer. We have corrected the typos that the reviewer pointed out.

Referee #3 (Comments on Novelty/Model System for Author):

This study has used comprehensive and complementary approaches including Pbk kinase dead knockin mice, Menin conditional knockout cell lines, JunD shRNA knockdown, inhibitors of Pbk, AP1, Menin, and HDAC, to demonstrate their points.

Referee #3 (Remarks for Author):

This manuscript addressed the role of Pbk, a serine/threonine protein kinase, in compensatory beta cell proliferation. Using many complementary approaches including Pbk kinase dead knockin mice, Menin conditional knockout cell lines, JunD shRNA knockdown, inhibitors of Pbk, AP1, Menin, and HDAC, this study gracefully demonstrated that (1) JunD first bound to the promoter of Pbk and recruited menin and HDAC3; (2) the JunD/menin/HDAC3 complex then reduced transcription-activating acetylated histone3 level at the Pbk promoter and the expression of Pbk proteins; and (3) high-fat diet increased Pdk+/insulin+ cells from nearly 0% to ~5% in the mouse islets. The manuscript was well-written with strong scientific rigor and high impact.

Response: We appreciate the positive feedback from the reviewer.

Minor concerns:

1. *The title needs to be revised because Pbk did not control menin/JunD/Pbk axis.*

Response: We have revised the title to the new one as "The menin/JunD/Pbk axis regulates compensatory beta cell proliferation" for better clarity.

2. *The data in Figure S1A and B are from public datasets, and need references.*

Response: We moved Figure S 1 to Appendix Figures in updated version of manuscript and included these two-public data in the section of Data citation in revised manuscript.

3. *It is important to document that the proliferating cells in the islet were Pbk+ in Figure 1D, 1G, and 8E. It helps define whether the compensatory beta cell proliferation effect of Pbk is cell-autonomous or not.*

Response: We performed co-staining of pancreatic sections with the Pbk antibody and the BrdU antibody to examine co-localization of Pbk and BrdU. The results showed co-localization of Pbk and BrdU and about 70% Pbk positive staining cells were also BrdU positive. We present these new data as follows here and also as new Fig1J and K and Fig 8 K in the revised manuscript. These results suggest that Pbk-mediated cell proliferation is cell-autonomous. We added the comments about the new data in Line 108-110 and line 309-310.

Fig. 5. (Updated Figure 1J and K, Figure 8K in revised manuscript). Pbk-mediated cell proliferation is cell-autonomous. (A) The representative image of co-staining of Pbk and BrdU in pancreatic sections. (B) Quantification of the cell percentage with pbk and BrdU co-staining in HFD-fed mouse islets. (C) Quantification of the cell percentage with pbk and BrdU co-staining in MI-treated mouse islets.

4. Why did ectopic expression of Pbk increase the sensitivity of cells to the Pbk inhibitor?

Response: Ectopic expression of Pbk led to enhanced growth of the cells (Fig. 1L). We think that the Pbk-expressing cells became more dependent on the ectopically expressed Pbk for maintaining the enhanced growth. As such, the cells with ectopic Pbk expression became more sensitive to the Pbk inhibitor. These results also suggest the specific role of Pbk inhibitor in repressing cell growth by targeting Pbk. In the revised manuscript, we moved the old Fig1L into Figure EV1A.

5. Why did PBK kinase dead knockin only impair glucose tolerance in male mice, but not in female mice? Can the authors provide any thoughts in discussion? Any correlation in human male vs female patients?

Response: It is not clear why male Pbk KI/KI mice develop IGT. Estrogen, which has a higher level in female, is pro-proliferative for beta cells (Choi *et al*, 2005; Le May *et al*, 2006), and protects beta cells from apoptosis through estrogen receptor alpha, and prevent type I diabetes in mice. Thus, the select effect of Pbk KI/KI on male mice may attribute to the higher level of estrogen in female mice. Relevant to this, it was reported that type 2 diabetes is more common among middle-aged men in populations of European extraction (Logue *et al*, 2011). To address the reviewer's question, we added the following sentences in Discussion section in line 378 to 384: "In our mouse model, we also found that male mice with PbkKI/KI selectively developed IGT, whereas female mice did not (data not shown). As elucidated by previous studies, estrogen is pro-proliferative for beta cells (Choi *et al.*, 2005; Le May *et al.*, 2006), and protects beta cells

from apoptosis through estrogen receptor alpha and prevent type I diabetes in mice (Le May *et al.*, 2006). Thus, the select effect of Pbk kinase dysfunction KI on male mice may be attributed to lower levels of estrogen compared to female mice.”

6. In the result section regarding Figure 2F-G, please specify that these were male mice.

Response: We have made this revision in line 151 as the reviewer suggested.

7. Unlike Fig. 4L, Fig. S5C did not show AP1 inhibitor induced Pbk expression in a dose-dependent matter (no induction at 0.1 and 1 μ M concentration). Please revise the result.

Response: We have deleted “in a dose-dependent matter” about the related description about Fig. 4L in the revised manuscript in line 231-232.

8. Can the authors provide any thoughts on how high fat diet affected the recruitment of JunD/menin/HDAC3 on the Pbk promoter?

Response: We further examined whether HFD affects the expression level of menin or HDAC3 in islets that may influence Pbk expression, however, we did not observe any changes in levels of these two proteins. These data were shown below:

Fig. 6. The expression level of menin and HDAC3 in islets from HFD or chow diet fed mice. (A) Men1 gene mRNA level in mouse islets detected by qRT-PCR. (B) HDAC3 gene level in mouse islets detected by qRT-PCR. (C) Western blot showed protein levels of menin, Pbk, and HDAC3 in islets from HFD or chow diet fed mice. n.s. no significance.

As it was reported that transcription factor FoxM1 is required for HFD-induced beta cell proliferation in mice (Golson *et al.*, 2010), it is possible that HFD may influence FoxM1 to modulate Pbk expression. On the other hand, it is also conceivable that HFD indirectly affects the recruitment of JunD/menin/HDAC3 to the Pbk promoter. These various scenarios warrant further investigation. To address the reviewer’s comment, we added the following sentences to the Discussion section in line 346-350: “Other possible mechanisms are certainly possible that warrant further investigation. It was reported that transcription factor FoxM1 is required for HFD-induced beta cell proliferation in mice (Golson *et al.*, 2010), therefore it is possible that HFD may

influence FoxM1 to modulate Pbk expression. On the other hand, it is also conceivable that HFD indirectly affects the recruitment of JunD/menin/HDAC3 to the Pbk promoter.”

9. *Can the authors provide explanation why there was no Pbk in lane 1 of Fig. 3C, but Pbk can be detected in lane 1 of Fig. 4B?*

Response: The variable exposure time for the film caused the difference in Pbk basal expression between Figure 3C and Figure 4B, as extensively detailed at #7 in response to reviewer 1. We have updated Figure 3C with a longer exposed image to avoid the confusion.

10. *In line 204, it should be Fig. 4D-F, not Fig. 1D-F.*

Response: This has been corrected as the reviewer suggested.

11. In line 314, it should be oncogenic, not ocogenic.

Response: The “Oncogenic” was deleted in rewrote sentence in line 326-327.

12. *Please verify the y-axis unit of the ChIP figures.*

Response: We have verified and revised the y-axis unit in the figure 5E, 5F, as detailed at #8 in response to reviewer 1.

Once again, we greatly appreciate the insightful and constructive comments and suggestions to improve our manuscript. We have revised the manuscript thoroughly, including performance of multiple lines of new experiments and addition of relevant clarifications and explanations, according to the insightful suggestions from the reviewers. We hope that you and the reviewers agree that the revised manuscript has been significantly improved and is now acceptable for publication.

Sincerely,

Xianxin Hua, MD, PhD

Professor of Cancer Biology
Department of Cancer biology
Institute for Diabetes, Obesity, and Metabolism,
University of Pennsylvania Perelman School of Medicine

REFERENCES:

- Abe Y, Matsumoto S, Kito K, Ueda N (2000) Cloning and expression of a novel MAPKK-like protein kinase, lymphokine-activated killer T-cell-originated protein kinase, specifically expressed in the testis and activated lymphoid cells. *J Biol Chem* 275: 21525-21531
- Abe Y, Takeuchi T, Kagawa-Miki L, Ueda N, Shigemoto K, Yasukawa M, Kito K (2007) A mitotic kinase TOPK enhances Cdk1/cyclin B1-dependent phosphorylation of PRC1 and promotes cytokinesis. *J Mol Biol* 370: 231-245
- Choi SB, Jang JS, Park S (2005) Estrogen and exercise may enhance beta-cell function and mass via insulin receptor substrate 2 induction in ovariectomized diabetic rats. *Endocrinology* 146: 4786-4794
- Gaudet S, Branton D, Lue RA (2000) Characterization of PDZ-binding kinase, a mitotic kinase. *Proc Natl Acad Sci U S A* 97: 5167-5172
- Golson ML, Misfeldt AA, Kopsombut UG, Petersen CP, Gannon M (2010) High Fat Diet Regulation of beta-Cell Proliferation and beta-Cell Mass. *Open Endocrinol J* 4
- Le May C, Chu K, Hu M, Ortega CS, Simpson ER, Korach KS, Tsai MJ, Mauvais-Jarvis F (2006) Estrogens protect pancreatic beta-cells from apoptosis and prevent insulin-deficient diabetes mellitus in mice. *Proc Natl Acad Sci U S A* 103: 9232-9237
- Logue J, Walker JJ, Colhoun HM, Leese GP, Lindsay RS, McKnight JA, Morris AD, Pearson DW, Petrie JR, Philip S *et al* (2011) Do men develop type 2 diabetes at lower body mass indices than women? *Diabetologia* 54: 3003-3006
- Matsumoto S, Abe Y, Fujibuchi T, Takeuchi T, Kito K, Ueda N, Shigemoto K, Gyo K (2004) Characterization of a MAPKK-like protein kinase TOPK. *Biochem Biophys Res Commun* 325: 997-1004
- Xing B, Ma J, Jiang Z, Feng Z, Ling S, Szigety K, Su W, Zhang L, Jia R, Sun Y *et al* (2019) GLP-1 signaling suppresses menin's transcriptional block by phosphorylation in beta cells. *J Cell Biol* 218: 855-870
- Zykova TA, Zhu F, Wang L, Li H, Bai R, Lim DY, Yao K, Bode AM, Dong Z (2017) The T-LAK Cell-originated Protein Kinase Signal Pathway Promotes Colorectal Cancer Metastasis. *EBioMedicine* 18: 73-82

3rd Feb 2021

Dear Prof. Hua,

Thank you for the submission of your revised manuscript to EMBO Molecular Medicine. I am pleased to inform you that we will be able to accept your manuscript pending the following final amendments:

- 1) Please address the referee suggestion.
- 2) Title: Please consider revising the title so that it reflects the main finding of the study. For example, "Pbk regulates high fat diet-induced compensatory beta cell proliferation". Avoid series of gene names, e.g. *menin/JunD/Pbk*.
- 3) In the main manuscript file, please do the following:
 - Correct/answer the track changes suggested by our data editors by working from the attached/uploaded document.
 - Add up to 5 keywords.
 - Remove text highlight color.
 - Add figure callouts for EV Fig4A-D.
 - Make sure that all special characters display well.
 - In M&M, include a statement that informed consent was obtained from all human subjects and that the experiments conformed to the principles set out in the WMA Declaration of Helsinki and the Department of Health and Human Services Belmont Report.
 - Rename Competing interest to Conflict of interest.
 - Indicate in legends exact $n=$ and exact $p=$ values, not a range, along with the statistical test used. To keep the figures "clear" some authors found providing an Appendix table Sx with all exact p -values preferable. You are welcome to do this if you want to.
- 4) Data availability: Data availability should contain only information about deposited data generated in this study. Please remove all information/links to deposited data published previously and cite the respective publication at an appropriate place in M&M/Results section. If no data generated in this study are deposited in external repositories, please state "This study includes no data deposited in external repositories." Please check "Author Guidelines" for more information. <https://www.embopress.org/page/journal/17574684/authorguide#availabilityofpublishedmaterial>
- 5) Appendix: Please remove suppl. M&M and add it to the M&M section in the main manuscript.
- 6) Funding: Please add HHS | NIH | National Cancer Institute (NCI) R01 DK097555 to X.H to Acknowledgements. Please make sure that information about all sources of funding are complete in both our submission system and in the manuscript.
- 7) For more information: There is space at the end of each article to list relevant web links for further consultation by our readers. Could you identify some relevant ones and provide such information as well? Some examples are patient associations, relevant databases, OMIM/proteins/genes links, author's websites, etc...
- 8) As part of the EMBO Publications transparent editorial process initiative (see our Editorial at <http://embomolmed.embopress.org/content/2/9/329>), EMBO Molecular Medicine will publish online a Review Process File (RPF) to accompany accepted manuscripts. This file will be published in conjunction with your paper and will include the anonymous referee reports, your point-by-point response and all pertinent correspondence relating to the manuscript. Let us know whether you agree with the publication of the RPF and as here, if you want to remove or not any figures from it prior to publication. Please note that the Authors checklist will be published at the end of the RPF.
- 9) Please provide a point-by-point letter INCLUDING my comments as well as the reviewer's reports

and your detailed responses (as Word file).

I look forward to reading a new revised version of your manuscript as soon as possible.

Yours sincerely,

Zeljko Durdevic

***** Reviewer's comments *****

Referee #1 (Comments on Novelty/Model System for Author):

most of my previous concerns have been addressed.

Referee #1 (Remarks for Author):

This is a much improved version of the manuscript by Ma et al and I congratulate the authors on a detailed, comprehensive rebuttal letter, with an impressive array of additional data. I believe the paper is now stronger and more robust.

All of my points have been addressed. The only final point I would suggest is that they add reference to point 6 (GTT improvement) to the discussion-a sentence should suffice.

The authors performed the requested editorial changes.

We are pleased to inform you that your manuscript is accepted for publication and is now being sent to our publisher to be included in the next available issue of EMBO Molecular Medicine.

Corresponding Author Name: Xianxin Hua
Journal Submitted to: EMBO Molecular Medicine
Manuscript Number: EMM-2020-13524